# Using a mixed opinion dynamics and innovation diffusion model to explore the 'best game no one played' phenomenon

Chung-Yuan Huang[1], Sheng-Wen Wang[2]*

1 Department of Computer Science and Information Engineering, Chang Gung University, Taoyuan City, Taiwan, Republic of China, 2 Department of Finance and Information, National Kaohsiung University of Science and Technology, Kaohsiung City, Taiwan, Republic of China

* swwang@nkust.edu.tw

## Abstract

Innovative products that receive favorable reviews but never catch on with consumers belong to a category known as the "best game no one played." We combined an adoption threshold model with an opinion dynamics model to examine reasons why certain high-quality products and ideas never achieve expected levels of commercial success. Computational social scientists use opinion dynamics models to analyze consensus formation, and adoption threshold models to study acceptance scenarios. However, most studies based on the first type focus on opinion exchanges without discussing follow-up actions, and most based on the second type only examine ways that individual decisions are dependent on numbers or proportions of friends and neighbors already engaged in specific behaviors, regardless of opinion differences. For this study, four kinds of theoretical networks (regular lattice, random, small-world, scale-free) served as underlying social network structures, and an agent-based simulation approach was used to analyze opinion exchange dynamics and product acceptance. Results indicate that computational agents were capable of changing pro/con opinions regarding issues, products, policies, etc. based on communication with neighboring agents via underlying social networks, and of making acceptance/ rejection choices based on a combination of individual adoption threshold plus observations of their neighbors' behaviors. A series of sensitivity analysis simulation experiments was conducted to identify model-related factors, determine non-linear correlations among them, and quantify degrees of influence. Factors exerting the strongest influence or requiring greater care when applied to cases of innovation diffusion were examined. Sensitivity analysis results indicate that agent adoption threshold mean exerted the greatest influence, followed by agent attitude mean and bounded confidence. Mechanism decomposition experiments revealed that the testimony effect neutralizes opinion clustering, making coordination failure the dominant driver of the opinion–adoption gap. These findings yield predictions distinguishing the model from information cascades, network externalities, and global games.

**Data availability statement:** All simulation source code (NetLogo 4.0.5 and Python 3 reimplementation) and parameter configuration files are publicly available without restriction under the MIT License at https://github.com/canslab1/BCAT, with a permanent archived copy deposited on Zenodo (DOI: 10.5281/zenodo.19216365). A step-by-step protocol for reproducing all simulation results is available on protocols.io (DOI: 10.17504/protocols.io.261geykydv47/v1). The minimal data set required to replicate all study findings is provided in four Supporting Information files accompanying the manuscript: S1 File (sensitivity analysis data, N = 100,548 observations), S2 File (mechanism decomposition data, 30,000 runs), S3 File (finite-size scaling data, N = 400 to 2,500), and S4 File (analysis scripts). These data are also available at https://github.com/canslab1/BCAT/tree/main/data.

**Funding:** The author(s) received no specific funding for this work.

**Competing interests:** The authors have declared that no competing interests exist.

## Introduction

Innovative products and ideas require careful pre-release planning and resource investment [1], with the potential for product acceptance and profitable sales generally increasing in step with the number of word-of-mouth recommendations [2–4]. However, due to the complexities of interpersonal communication and influence, even the most careful planning cannot ensure product success [5–10]. For offline friend/acquaintance networks and online social networks, the large majority of interactions occur between individuals who already know each other [11,12]. Depending on timing and attitude strength, opinion exchanges among social network friends may be sufficient for altering an individual's original views, sometimes even reversing prejudices. Possible results include public conformity and private acceptance [13,14].

Product acceptance and sales success involve innovation diffusion processes and adoption thresholds that have been studied by researchers in computer science, mass communication, and social psychology, among other topic areas (see, for example, [7,15–18]). Initially created by the American communication theorist and sociologist E. M. Rogers [19], diffusion of innovation theory (sometimes called "adoption of innovation") originally addressed the evolution of new concepts, products, trends, and technologies introduced to social systems. In *Diffusion of Innovations* [7] he listed four elements influencing the spread of new ideas—the innovation itself, communication channels, time, and the social system involved—and described dissemination as consisting of awareness, persuasion, decision making, and confirmation. The innovation diffusion literature contains many examples of large numbers of individuals wanting to participate in discussions about new concepts, products, trends, and technologies, yet widespread discussion does not guarantee acceptance. Lack of familiarity and certainty generally results in mixed levels of support based on factors such as cultural, religious, and political beliefs [7,20], social backgrounds, and social interdependence [21–23]. It is important to remember that opinion formation gradually and continually evolves as opinions change in response to environmental influences [24–26]. Further, a positive attitude does not guarantee action or success beyond a few pioneers willing to accept unpredictable consequences associated with being the first to try something [7,12]. Friends and acquaintances generally observe pioneers before making acceptance/rejection decisions [27,28], with more conservative individuals waiting until visible majorities test new products or ideas. But in the end, strong sales or broad approval can never be guaranteed [7].

There are numerous examples of products and ideas that were considered innovative when released, but never caught on with consumers—examples of "the best game no one played" phenomenon associated with diffusion of innovation [29]. The Palm hand-held computer, released in 1996, was immediately adopted by a small number of tech-savvy individuals, but never reached the level of popularity its designers had hoped for [30]. Another example comes from Taiwan, where the film "The Bold, The Corrupt and The Beautiful" won Best Film and Best Drama audience awards at the 54th Golden Horse Film Festival, but failed at the box office [31]. A third example is the "service-oriented architecture" (SOA) concept, a software design feature in which services were provided via network communication protocols. It was

perceived as having great benefit for organizations when introduced in 2005, but its high entry barriers and short-term benefits were difficult to overcome. Resource investment in SOA dried up during the 2008 economic downturn.

Diffusion of innovation studies are found in multiple topic areas. The social psychologists Martin Fishbein and Icek Ajzen [32] applied rational choice theory to discuss ways that individual decision-making behaviors are affected by internal ideas and external pressures. Ajzen's [33] theory of planned behavior, an expansion of rational choice theory, contains an additional personal willpower factor. Another expansion of rational choice theory, Davis et al.'s [34] technology acceptance model, addresses ease of use and ways that usability can influence individual decisions to adopt or reject new technology products. In the field of communications, Oliver et al.'s [35] critical mass theory explains how diffusion speed increases and promotion becomes easier once the number of group pioneers reaches a critical mass.

However, social science research requires large amounts of data collected over long time periods, using questionnaires designed to gather information on inner thoughts and feelings involving specific decision-making behaviors within limited time frames. Since individuals tend to continually adjust their attitudes based on opinion exchanges [25], data gathered with questionnaires and other time point collection methods often fail to accurately capture opinion dynamics. To address this weakness, researchers such as Deffuant et al. [36], Hegselmann and Krause [6] and Sznajd-Weron and Sznajd [37] have designed agent-based and network-oriented simulation approaches to measure the effects of various factors on decision-making behaviors. Further, many information science researchers employ opinion dynamics models of ongoing intra-group interactions and opinion exchanges to analyze consensus evolution [6]. Such models have been applied in studies of mainstream media consumption [38], administrative policies [39], biological organisms [40], microfinance loans [41], and corporate leadership [42,43], among many other topics [44–46].

The primary assumption underlying adoption threshold models of innovation diffusion is that individual actions are dependent on the number of adjacent or nearby individuals doing the same [10,12]. Karimi and Holme [16] used a version of this model to examine how chronological sequences of events and contact periods affect temporal networks, hypothesizing that the timing of interaction and idea infection processes are affected by contacts within specific time frames. Backlund et al. [47] added randomness and certainty to Karimi and Holme's temporal network, and found that the number of contacts between individuals and neighbors during specified time periods affects purchase decisions—that is, when individuals come into contact with multiple neighbors who purchase a product, they are more likely to do the same.

Some computational social science researchers have experimented with both opinion dynamics and adoption threshold diffusion of innovation models to examine opinion exchanges and consensus formation. However, many studies using models in the first category emphasize opinion exchanges without discussing follow-up actions, and many using models in the second category only address individual decisions that are completely dependent on proportions of friends and neighbors already engaged in a specific behavior, regardless of opinion differences. To our knowledge, no effort has been made to combine the two model types for purposes of simulating opinion exchanges and observing their effects on decisions and actions. We believe such a combination can support a broader understanding of opinion exchanges and product acceptance.

Our study goals are to establish a more complete picture of diffusion of innovation and opinion evolution processes, and to examine the results of their interaction. This model-merging effort involves four social network models: regular lattice (CA, two-dimensional toroidal cellular automata with Moore neighborhood and periodic boundary conditions), random (RN), small-world (SWN, with 0 < rewiring rate < 1), and scale-free (SFN). We assumed positive/negative polarity in attitudes on controversial issues, with individuals who hold positive attitudes more likely to accept and purchase innovative products. One advantage of combining opinion dynamics and adoption threshold models is that the first considers the evolution of opinions in social networks over time, and the second addresses factors that affect actions and decisions. Sensitivity analysis simulation experiments were conducted to identify primary model-related factors, determine non-linear correlations among them, and quantify degrees of influence. We investigated which factors exert the greatest influence, and which ones require greater care when applied to cases of innovation diffusion.

## Related opinion dynamics and adoption threshold models

Opinion dynamics models simulate opinion formation following exposure to multiple information sources such as social media and observations of neighbor actions. Their purposes are to identify opinion exchange details, speculate on how best to achieve public consensus, determine reasons for opinion polarization, and predict the effects of adding small numbers of outlier opinions. Random values are assigned to individual agents during model initialization, each indicating an original opinion. Opinions are usually coded as real numbers with ranges known as *opinion spaces*, continuous in some models and discrete in others. In social network models, agent relationships are expressed as links through which opinions are exchanged. The main feature of Friedkin and Johnsen's [48] *bounded confidence* model is that exchanges are limited to individuals who have similar opinions—in their words, communication only occurs when opinion differences fall below a trust boundary. One bounded confidence model extension, Hegselmann and Krause's [6] *HK model*, measures opinions as real numbers with ranges of [0, 1]. The most important HK model parameter is uncertainty $\varepsilon$, a trust boundary sometimes referred to as "controversy tolerance." During opinion exchanges, randomly selected agents treat the averages of all opinions within $\varepsilon$ ranges as new opinions. During simulations, opinion exchange iterations are repeated until a defined opinion distribution is achieved.

A key task in HK model simulations is controlling parameters $\varepsilon_l$ and $\varepsilon_r$, whose values are reset for each generation when addressing asymmetric cases. When an agent's opinion is closer to 1, $\varepsilon_r > \varepsilon_l$; when closer to 0, $\varepsilon_l > \varepsilon_r$. The HK model lacks a social network structure, therefore agents exchange opinions as long as the opinion difference falls within a specified $\varepsilon$ value range. Due to its characteristic of making full use of bounded confidence features, researchers have applied various HK model versions to examine microfinance loans [41], mainstream media [38], administrative agency policies [39], in vivo bioactivity [40], and corporate leadership [42,43], among other topics.

In contrast, Deffuant et al.'s [36] relative agreement model (sometimes referred to as the RA or D model) considers the circumstances in which extreme opinions evolve into social norms that affect all individuals. Both D and HK models are based on bounded confidence, but the D model opinion space parameter is set to [−1, +1]. The most important difference between the D and HK models is that D model opinion exchanges are bidirectional, meaning that two parties can influence each other's opinions. During our experiments, agent pairs were randomly selected and checked to determine whether their opinion distances were below the $\varepsilon$ parameter. Agent opinions did not change when distances were equal to or above $\varepsilon$. Opinion value adjustments were based on Eqs (1) and (2),

$$x_j = x_j + u_i \left( \frac{h_{ij}}{u_i} - 1 \right) (x_i - x_j)$$

(1)

$$u_j = u_j + u_i \left( \frac{h_{ij}}{u_i} - 1 \right) (u_i - u_j)$$

(2)

where $x_i$, $x_j$ and $u_i$, $u_j$ are the opinions and uncertainties of agents $i$ and $j$, respectively, and $h_{ij}$ is their overlapping confidence interval, expressed as $[x - u, \ x + u]$.

The non-overlapping length for agent $j$ is $2u_i - h_{ij}$. The value of an overlapping length minus non-overlapping length is called *agreement*—the greater the agreement value for two agents, the more consistent their opinions. The relative consistency of agent $j$ is defined as the agreement value divided by the length of agent $i$'s confidence interval. Accordingly, the relative consistency of agent $j$ to agent $i$ is $[h_{ij} - (2u_i - h_{ij})]/2u_i$ (equal to $h_{ij}/u_i - 1$), and the relative consistency of agent $i$ to agent $j$ is $[h_{ij} - (2u_j - h_{ij})]/2u_j$ (equal to $h_{ij}/u_j - 1$). Changes in opinions are determined as $\mu \times$ (relative consistency) $\times$ (opinion difference between agents $i$ and $j$). Undecided individuals are more likely to change than individuals who are more confident in their decisions—that is, they are more likely to be affected by opinion exchanges.

As stated, agent opinion exchanges in the D model are bidirectional. Another significant difference between the D and HK models is that the uncertainty value of the first varies in step with agent opinion exchanges. The D model focuses on differences between extreme and moderate opinions: extreme agents are those with exceptionally low uncertainty, and whose opinions are close to the opinion space boundary; moderate agents are those with uncertainty and opinion values within one SD of their respective mean values. The HK model assumes agent opinion homogeneity. Our proposed model does not implement the full RA updating rule described in Eqs (1) and (2), but employs a simplified fixed-rate pairwise updating mechanism as detailed in Table 1 and the "Simulation model specifications" section below.

This mechanism is based on the bounded confidence tradition established by the two frameworks, but stands as a distinct variant sharing features with both. Each agent randomly selects one neighbor per time step, following the pairwise random interaction approach of Deffuant [49] rather than the synchronous all-neighbor aggregation of the HK model. The bounded confidence threshold is a globally fixed parameter shared by all agents throughout the simulation. It does not vary by agent or over time, as it does in the Deffuant et al. (2002) [36] Relative Agreement model. Attitude value integers are in [1, 100], thus forming a uniform 100-level discretization of the continuous opinion space [0, 1], with each integer unit corresponding to an interval of 0.01. This precise mapping preserves all qualitative continuous-space dynamics properties. The convergence rate $\mu$, which governs attitude adjustment magnitude, plays a role analogous to the $\mu$ parameter in the Deffuant framework, but without the relative-agreement weighting that scales adjustment according to the ratio of confidence interval overlap to uncertainty width. Table 1 summarizes relationships among the HK model, the Deffuant Relative Agreement model, and the opinion update mechanism of our proposed model.

Adoption threshold models address the assumption that the actions and decisions (or lack of) of an individual depend on the number and proximity of surrounding friends/acquaintances who take the same actions or make the same decisions [10]. Most individuals refrain from trying new products due to the perceived unpredictability of consequences, and therefore rely on their observations of the reactions of friends when making their own purchase decisions. Once the number of approving friends exceeds a defined adoption threshold value, the odds of making the same decision or taking the same action increase significantly. Compliant individuals with lower adoption thresholds are more likely to take the same actions as their friends.

In network-oriented adoption threshold models, two factors determine when agents (nodes) adopt the opinions of friends: adoption threshold value and number of neighbors. In the example shown in Fig 1, Andy's adoption threshold is 1/3, meaning that when more than 1/3 of his neighbors or friends adopt a new product or opinion, Andy will follow suit. There are two situations that make it easy for individuals to perform actions. In the first, an agent's adoption threshold is very low, meaning that he is easily influenced by friends. In the second the number of neighbors is very small, therefore

**Table 1. Structural comparison of the HK model, the Deffuant Relative Agreement (RA) model, and the opinion update mechanism of the model proposed in this study. This opinion component follows the pairwise random interaction structure of the Deffuant approach with a fixed global confidence threshold, applied on an explicit network topology and conditioned on agents' adoption statuses. The integer attitude scale [1, 100] is a uniform discretization of [0, 1] (step size = 0.01) and is mathematically equivalent to the continuous-space Deffuant formulation.**

| Model Feature | HK Model | Deffuant RA Model | Proposed Model |
|---|---|---|---|
| Interaction structure | All neighbors within $\varepsilon$ simultaneously | Single randomly selected neighbor | Single randomly selected network neighbor |
| Confidence threshold | Fixed, uniform $\varepsilon$ | Agent-specific, time-varying $u_i$ | Fixed, uniform $\varepsilon$ |
| Opinion space | Continuous [0, 1] | Continuous [−1, +1] | Discrete integer [1, 100] ≡ [0.01, 1.00] |
| Update timing | Synchronous | Sequential pairwise | Pseudo-concurrent (*ask-concurrent*) |
| Uncertainty dynamics | None | $u_i$ updates after each interaction | None |
| Convergence parameter | Not explicit | $\mu$ (fixed) | $\mu$ (fixed) |
| Network structure | None (fully mixed) | None (fully mixed) | Explicit network topology |
| Adoption-status conditioning | Absent | Absent | Present (4 scenarios) |

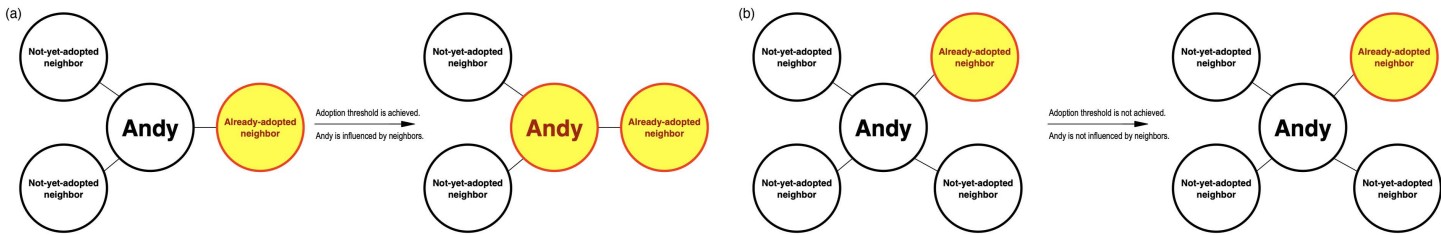

**Fig 1. Schematic diagram of the adoption threshold model used in this study.** Black-framed circles indicate not-yet-adopted neighboring agents, red-framed circles already-adopted neighboring agents. Andy makes his decision based on the actions of his neighboring agents. **(a)** One of Andy's three neighbors has acted, thus meeting the required 0.3 adoption threshold. **(b)** In this case Andy has four neighbors, of which only one has adopted the product, therefore the 0.3 adoption threshold is not met.

individual neighbors can wield significant influence with an agent—any action taken by a neighbor increases the likelihood that the agent will do the same. Andy has three neighbors in the Fig 1a example, and if only one takes an action, a 1/3 adoption threshold is immediately achieved. Andy's adoption threshold is also 1/3 in the Fig 1b example, but since there are four neighbors, each one exerts a smaller impact.

Prior applications of adoption threshold models to analyze innovation diffusion have generally been limited to discussing numbers of nearby agent friends and their influences on agent decision-making, without considering the inner attitudes of agents toward products. Assuming that all individuals have opinions on different topics, we based our model on the one described in Wang et al. [13,14] and Hsieh [50] to address controversial issue diffusion in opinion dynamics and adoption threshold contexts. One resulting characteristic is that our proposed model can use explicit behaviors to predict agent opinions. Once an agent adopts a new product, our model deduces the positive attitudes of many of the agents' friends who also use it. However, this is a one-way process: just because an agent has a positive attitude toward a product (and has many friends who also approve of it) does not guarantee that the agent will purchase and use it—such decisions are based on whether or not an adoption threshold is exceeded. Our model assumes that even though an agent's opinion of a new product may be hidden, it is still possible to predict the agent's future action based on how many of its friends are using it.

For theoretical reasons tied to the adoption mechanism introduced above, bounded confidence dynamics was chosen as our model's primary opinion formation component. Alternative opinion update rules—especially DeGroot-style weighted averaging [51–53] and voter model dynamics—lack the property that makes bounded confidence a natural complement to threshold-based adoption. According to DeGroot updating in a strongly connected network, repeated weighted averaging drives all agents toward a single consensus opinion determined by the structure of the network's influence centrality, thus making the long-run opinion distribution homogeneous. Accordingly, any adoption failure in a DeGroot-based model must arise from threshold barriers acting on a population that has already converged to a shared attitude. Opinion fragmentation cannot contribute to adoption outcomes because the model rules it out by design. The voter model similarly converges to consensus states while operating on discrete opinion values. Neither property is suitable for representing the graded attitude intensities and selective communication boundaries that reflect the well-documented homophily found in social networks [54].

In contrast, the bounded confidence mechanism produces stable and persistent opinion clusters when defining long-run outcomes involving subpopulations that have converged to local equilibria and stopped communicating across cluster boundaries because their mutual attitude distances exceed the threshold. This property has a direct and important consequence for adoption dynamics. In our proposed model, adoption eligibility requires a strictly positive attitude above the opinion scale midpoint. Any opinion cluster with an equilibrium at or below this midpoint permanently excludes its members from adopting a product or idea, regardless of their local network structure or threshold value. The bounded

confidence mechanism thus acts as a built-in initial filter on the potential adopter population, creating a wedge between the proportion of agents who hold favorable opinions and the proportion who are eligible to enter the adoption decision process. Neither DeGroot averaging nor voter dynamics are capable of producing this wedge as an emergent equilibrium property.

## Simulation model specifications

To explore the best game no one played phenomenon, we integrated an established bounded confidence-based opinion dynamics model with an equally well-tested adoption threshold innovation diffusion model to construct novel agent-based, network-oriented simulations addressing specific issues. We used the initials for bounded confidence and adoption threshold to name the model BCAT, and applied it to identifying factors associated with good sales probabilities, non-linear correlations among them, factors exerting the greatest influence, and factors requiring greater care when applied to instances of innovation diffusion. Our goal was to help policy researchers, government officials, and marketers adjust parameters to create functional scenarios for studying opinion dynamics, to test adoption diffusion processes involving novel products or controversial policies, and to investigate marketing and promotional strategy effectiveness. In the rest of this section we will describe BCAT's computational implementation process, agent initialization procedure, and algorithmic rules, then discuss evaluation indicators used to quantify simulation outcomes as well as properties arising from coupling the two component mechanisms.

The BCAT user interface is shown in Fig 2, simulation process flowchart in Fig 3, and pre-simulation specification parameters in Table 2. Original model implementation, entailing Wilensky's [55] NetLogo agent-based modeling and simulation platform [56], was revised using Python 3 to create a new graphical interface facilitating reproducibility. Both implementations are available at https://github.com/canslab1/BCAT (MIT License), with a permanent archived copy on Zenodo (DOI: 10.5281/zenodo.19216365). A step-by-step protocol for reproducing all simulation results is available on protocols. io (DOI: 10.17504/protocols.io.261geykydv47/v1). To achieve statistical significance and control experimental error, each simulation was repeated 1,000 times using the same initial conditions and parameter settings [57]. As shown in Fig 3, each run was reset to an initial environment and performed for at least 300 simulation ticks. One iteration between an individual agent and a randomly chosen neighboring agent occurred per tick. Statistical plots reflect real time simulations until all factors of interest either achieved a steady state or met termination criteria. After each simulation run, all relevant initial conditions, parameter settings, and experimental outcomes were stored as external csv format files until further comparisons and sensitivity analyses were performed (Fig 3, Phase 6).

BCAT characterizes abstract forms of societal structures as theoretical networks in which nodes represent individuals (agents) and edges connections between individuals (friendships, family members, bosses, coworkers, etc.). As shown in the Fig 2 visual network model of the graphical user interface and the Algorithm 1 network model pseudo-code, underlying social networks consisting of 400 nodes and 1600 edges correspond to four kinds of theoretical networks: regular lattice, random, small-world and scale-free. Note that each network node has an average of 8 neighboring agents.

## Algorithm 1. Pascal-style pseudo-code for network model construction.

```
; Establish social network connections according to selected topology.
procedure Create social network model (network type, rewiring probability) defined as
  ; network type is a custom enumeration data type defined as a network connection type.
  ; network type = (scale-free, regular lattice, small-world, random)

  ; rewiring probability is a tunable probability determining degree of randomness
  ;      of the generated network model.

  if (network type is scale free) then
    Create a scale-free social network model G(N,E) using a preferential attachment growth
algorithm.
```

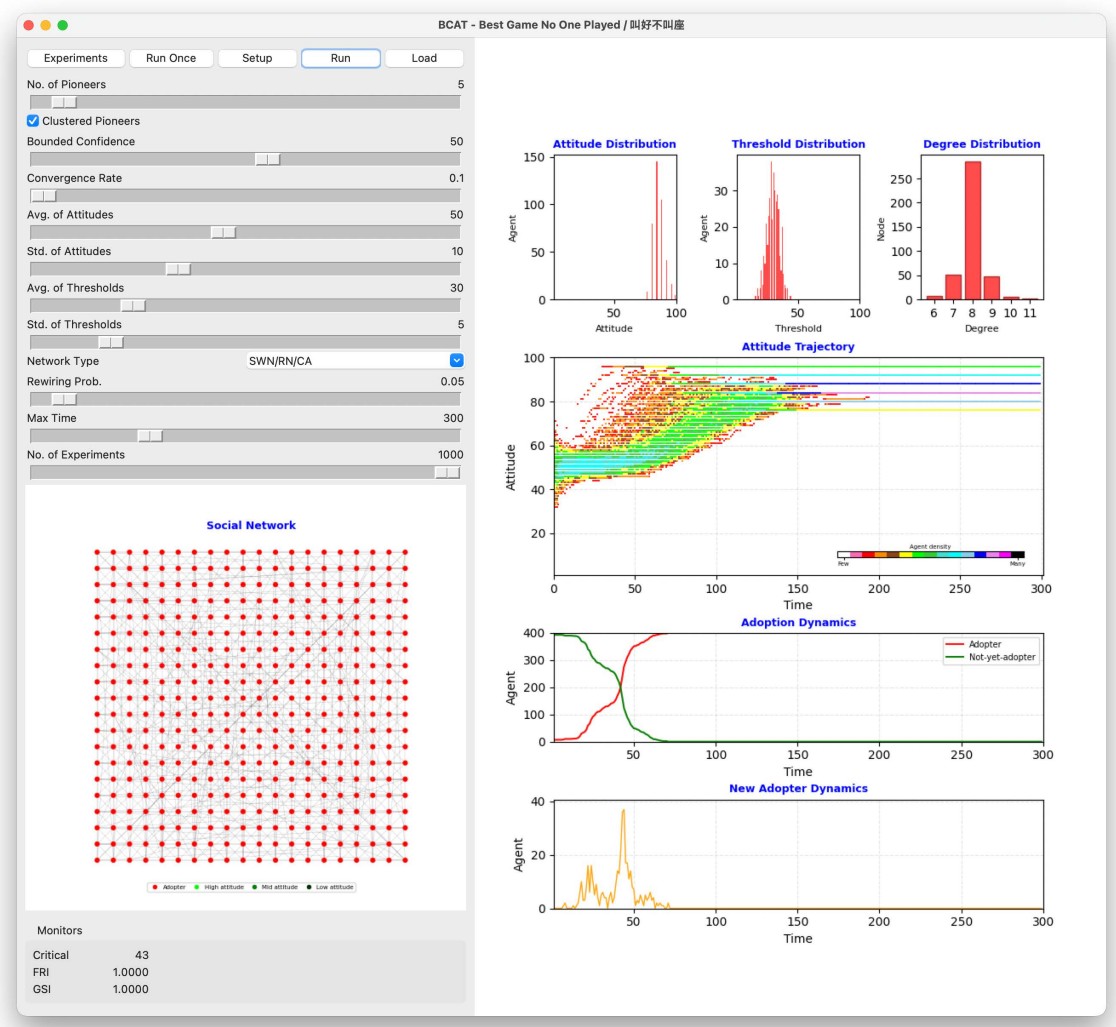

**Fig 2. Graphical user interface of the Python 3 BCAT simulation.** Parameter setting area is along the left side. Upper right: attitude, threshold, and degree distribution plots. Middle right: network agent attitude trajectory over time. Pink, red, orange, brown, yellow, green, lime, turquoise, cyan, sky, blue, violet, magenta and black colors indicate low-to-high agent numbers. Lower right: adoption dynamics (red S-shaped curve represents cumulative adoption over time, green inverse S-curve cumulative non-adoption over time) and new adopter dynamics (orange bell-shaped curve). Lower left: social network structure arranged in a two-dimensional grid. Red node represents already-adopted agent, green not-yet-adopted agent. Darker color indicates higher attitude value.

```
else
   Create a toroidal cellular automata G(N,E) with Moore neighborhood and periodic boundary
conditions.
   if (user-specified rewiring probability!=0) then
      Randomly rewire each edge of G(N,E) with a user-specified rewiring probability.
Arrange all network nodes in a two-dimensional grid.
```

In addition to scale-free networks created with a Barabási and Albert [58] preferential attachment growth algorithm, BCAT uses the widely recognized Watts and Strogatz [59] small-world network construction algorithm (with rewiring

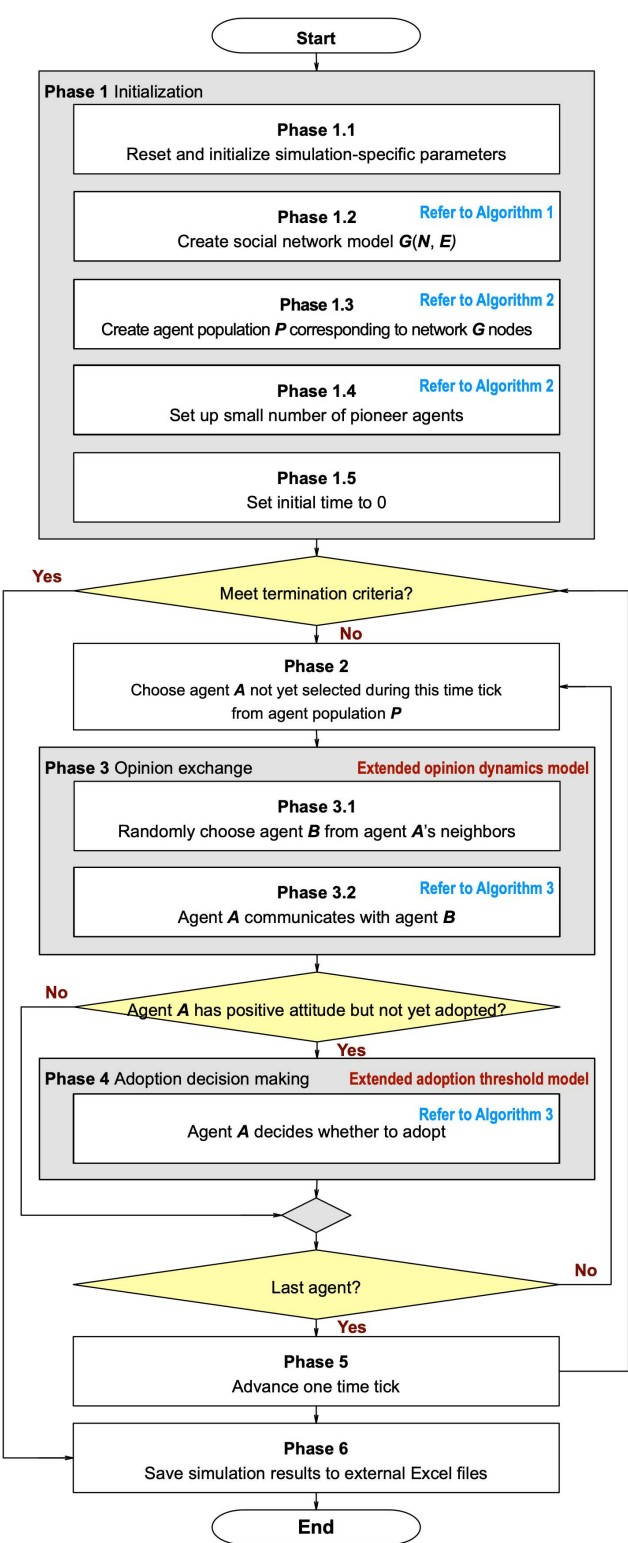

**Fig 3. Simulation process flowchart for proposed BCAT model.** Simulation process time complexity is $O(NT)$, where $N$ and $T$ respectively represent the number of agents and number of simulation time steps.

**Table 2. BCAT model and simulation system parameters.**

| Parameter | Description |
|---|---|
| *no-of-pioneers* | Number of pioneer agents with highest positive attitude values, therefore first to adopt an idea or purchase a product. Refer to **chosen-leaders** function in Algorithm 2 for details on parameter usage. |
| *clustered-pioneers?* | Indicating whether pioneer agents are concentrated within specific network nodes (= ON) or randomly scattered throughout a social network (= OFF). Refer to **chosen-leaders** function in Algorithm 2 for details on parameter usage. |
| *bounded-confidence* | Communicable range indicating agent tolerance of differences in neighboring agent attitudes—the larger the range, the greater the tolerance. Refer to **opinion-exchange** procedure in Algorithm 3 for details on parameter usage. |
| *convergence-rate* | Attitude ratio for two agents moving toward each other. Refer to **one-way-persuasion-process** and **two-way-communication-process** procedures in Algorithm 3 for details on parameter usage. |
| *avg-of-attitudes*, *std-of-attitudes* | Two parameters used together to set all agent attitudes during simulation process Phase 1.3 (Fig 3). NetLogo's **random-normal *mean stdev*** command is used to set each agent's attitude (*att*) to a normally distributed random integer between 1 and 100, with a mean (*avg-of-attitudes*) and standard deviation (*std-of-attitudes*). The integer attitude scale [1, 100] represents a uniform 100-level discretization of the continuous opinion space [0, 1] used in the bounded confidence literature, where each integer unit corresponds to an interval of 0.01. The midpoint (*att* = 50) corresponds to an opinion value of 0.50, separating negative attitudes (*att* ≤ 50) from positive attitudes (*att* > 50). Refer to **setup-agent-population** procedure in Algorithm 2 for details on parameter usage. |
| *avg-of-thresholds*, *std-of-thresholds* | Two parameters used together to set all agent adoption thresholds during simulation process Phase 1.3 (Fig 3). NetLogo's **random-normal *mean stdev*** command is used to set each agent's adoption threshold (*theta*) to a normally distributed random integer between 1 and 100, with a mean (*avg-of-thresholds*) and standard deviation (*std-of-thresholds*). Threshold values follow the same [1, 100] ≡ [0.01, 1.00] discretization convention as attitude values: in the adoption decision rule (Algorithm 3), the integer threshold is divided by 100 before comparison with the neighbor adoption proportion, yielding fractional thresholds consistent with the standard convention in the diffusion literature [10,12]. For example, *avg-of-thresholds* = 40 corresponds to a mean fractional threshold of $\theta = 0.40$, requiring at least 40% of an agent's neighbors to have adopted before the agent itself adopts. Refer to **setup-agent-population** procedure in Algorithm 2 for details on parameter usage. |
| *network-type* | During simulation procedure Phase 1.2 in Algorithm 1 (Fig 3), **Create social network model** is used to construct a network specified by this parameter. The four network connection types are SFN (scale-free), CA (regular lattice), SWN (small-world) and RN (random). |
| *rewiring-probability* | Probability determining degree of randomness for the generated network model. |
| *max-time* | Maximum number of time ticks for a single simulation. |
| *no-of-experiments* | Number of experiment repetitions. |

probability as its primary parameter) to create the other three networks. When the user-specified rewiring probability = 0, the network construction algorithm creates a toroidal regular lattice with Moore neighborhood and periodic boundary conditions. The resulting network exhibits high degrees of both clustering and separation. According to network theory, the opposite of a completely ordered regular lattice is a random network. When the user-specified rewiring probability = 1, all network edges are rewired randomly. Our network construction algorithm produces random networks that lack clustering but have low degrees of separation between nodes. When the user-specified rewiring probability is between 0 and 1, the network construction algorithm creates a network exhibiting the small-world characteristics of high degree of clustering and low degree of separation. For ease of visual presentation, we arranged the 400 network nodes connected by 1600 edges as a two-dimensional 20 × 20 grid.

As shown in Algorithm 2, each BCAT agent has three primary attributes: attitude (*att*), adoption threshold (*theta*), and action (*act*). Attitude is defined as an agent's summary evaluation of an external issue, product, or idea with some degree of favor or disfavor. Exposure rate is the proportion of adopters in an agent's personal network at a given time. Since adoption threshold is the proportion of adopters in an agent's personal network, it represents the time-of-adoption exposure rate. From another perspective, adoption threshold refers to agent conformity: when the proportion of neighboring

agents accepting or adopting an issue, product, or idea exceeds this threshold, the agent also adopts it. Action is expressed as a binary Boolean value indicating whether an agent has performed an action or purchased a new product.

**Algorithm 2. NetLogo-style pseudo-code for pioneer agent initialization.**

```
; Each agent has three required primary attributes:
;      attitude (att), adoption threshold (theta), action (act).
; Attitude (att), an integer between 0 and 100, is defined as an agent's negative or positive eval-
uation of
;      an issue, product or idea.
; Adoption threshold (theta), an integer between 0 and 100, is defined as the proportion of adopters
within
;      a social group required for an agent to adopt the issue, product or idea.
; Action (act) is a Boolean attribute indicating whether an agent has performed an action or
;      purchased a new product.
turtles-own [att theta act......]; see NetLogo source code for other attributes, declarations and
settings.

; Phase 1.3: Create agent population corresponding to network nodes
; Initialize each agent's required primary attributes.
to setup-agent-population
  ; Each agent's att is set to a normally distributed random integer between 1 and 100 with
  ;       an avg-of-attitudes mean and std-of-attitudes standard deviation.
  ; Each agent's theta is set to a normally distributed random integer between 1 and 100 with
  ;       an avg-of-thresholds mean and std-of-thresholds standard deviation.
  ; avg-of-attitudes, std-of-attitudes, avg-of-thresholds, std-of-thresholds, cluster-pioneers? and
  ;       no-of-pioneers are from the parameter setting area of the graphical user interface (Fig 2).
  ask-concurrent turtles [
     set att max list 1 (min list 100 random-normal avg-of-attitudes std-of-attitudes); [1, 100]
     set theta max list 1 (min list 100 random-normal avg-of-thresholds std-of-thresholds); [1, 100]
     set act false
  ]
  ; Phase 1.4: Set up no-of-pioneers for pioneer agents
  ; Each pioneer's attitude (att), adoption threshold (theta) and action (act) are respectively
  ;       set to 100, 0 and true.
  ask-concurrent chosen-leaders [
          set att 100; highest positive attitude
          set theta 0; no adoption threshold
          set act true; has already performed an action or purchased a new product
  ]
end

; A small number of agents are selected as opinion leaders and pioneers.
to-report chosen-leaders
   report ifelse-value (clustered-pioneers? = true)
                           [max-n-of no-of-pioneers turtles [xcor + ycor]]
                             [n-of no-of-pioneers turtles]
end
```

As shown in Fig 3, agent populations must be created and three primary attributes initialized during phase 1.3 of each simulation run. Agent attitude is set to a normally distributed random integer between 1 and 100, *avg-of-attitudes* and *std-of-attitudes* derived from the parameter setting area of the BCAT user interface (Fig 2). Similarly, each agent's adoption threshold is set to a normally distributed random integer between 1 and 100 with an *avg-of-thresholds* and

*std-of-thresholds*. An agent's adoption threshold does not change after initialization, but attitude varies over time on a scale of 1–100. Values between 1 and 50 indicate negative attitudes toward new issues (1 = strongest disagreement), and values between 51 and 100 indicate positive attitudes (100 = strongest agreement).

After initializing the entire agent population, small numbers of agents are selected as opinion leaders and adoption pioneers, based on their having the highest degrees of positive attitude but not achieving the required thresholds for triggering idea adoption or product purchase. Pioneer attitude, adoption threshold, and action values are respectively set to 100, 0 and "true." To determine the aggregate effects and impacts of pioneers in different clusters on simulation results, the BCAT model allows for concentrations of selected pioneers in specific interconnected network node clusters during initialization phase 1.4 (Fig 3), as well as during dispersal throughout a social network according to the user's specific requirements.

During each opinion exchange phase, individual agents randomly select single neighboring agents for potential communication, which occurs only if the difference between the attitude values of the two agents falls below the *bounded-confidence* parameter that governs opinion exchange eligibility. This condition is equivalent to $|x_i - x_j| < \varepsilon$, where $\varepsilon$ = *bounded-confidence* / 100 , which is consistent with the standard Deffuant bounded confidence literature [49]. One of four interaction scenarios is applied when this condition is met, depending on the adoption statuses of both agents as specified in Algorithm 3. Attitude adjustment in each scenario follows a fixed-rate convergence rule: the attitude of the adjusting agent moves toward its partner's attitude according to the *convergence-rate* ($\mu$) of their difference: $att_{new}$ = round($C + \mu \times (B - C)$), where $C$ is the agent's current attitude and $B$ its neighbor's attitude. Depending on the scenario, BCAT applies this rule in one direction (persuasion) or both directions (communication). For all simulations described in this paper, the *convergence-rate* parameter was set to 0.1. The proposed model does not use an agent-specific uncertainty parameter—the bounded confidence threshold is a fixed global property of the simulation rather than a varying individual attribute.

**Algorithm 3. NetLogo-style pseudo-code for opinion exchange and adoption decision making.**

```
; Phase 3.2: Agent A communicates with neighboring agent B
to opinion-exchange [Neighbor's_act Agent's_act Neighbor's_att Agent's_att]

  ; A neighboring agent has already adopted an idea or product, but the agent has not.
  if (Neighbor's_act=true and Agent's_act=false and abs(Agent's_att - Neighbor's_att) <bounded-confidence) [
    ; If neighboring agent's attitude is more positive than the agent's
    ;     because the neighboring agent has already adopted an idea or product, then the agent
becomes more positive.
    ; Otherwise,
    ;     both agents adjust their attitudes.
    ifelse (Neighbor's_att>Agent's_att)
      [one-way-persuasion-process-1      Neighbor's_att Agent's_att]; Scenario #1.1: User-friendly
testimony
      [two-way-communication-process      Neighbor's_att Agent's_att]; Scenario #1.2
  ]

  ; An agent has adopted an idea or product but its neighboring agent has not.
  if (Neighbor's_act!=true and Agent's_act=true and abs(Agent's_att - Neighbor's_att) <bounded-
confidence) [
    ; If the neighboring agent's attitude is more positive than the agent's,
    ;     then both agents adjust their attitudes.
    ; Otherwise,
    ;     neighboring agent's attitude becomes more positive.
    ifelse (Neighbor's_att>Agent's_att)
      [two-way-communication-process      Neighbor's_att Agent's_att]; Scenario #2.1
      [one-way-persuasion-process-2      Neighbor's_att Agent's_att]; Scenario #2.2: User-friendly
testimony
  ]
```

```
  ; Both an agent and its neighboring agent have already adopted an idea or product.
  if (Neighbor's_act=true and Agent's_act=true and abs(Agent's_att - Neighbor's_att) <bounded-confidence) [
    ; If the neighboring agent's attitude is more positive than the agent's,
    ;     then the agent's attitude becomes more positive.
    ; Otherwise,
    ;     neighboring agent's attitude becomes more positive.
  ifelse (Neighbor's_att>Agent's_att)
    [one-way-persuasion-process-1    Neighbor's_att Agent's_att]; Scenario #3.1
    [one-way-persuasion-process-2    Neighbor's_att Agent's_att]; Scenario #3.2
  ]

  ; Neither the agent nor its neighboring agent has adopted an idea or product, and both agents
adjust their attitudes.
  if (Neighbor's_act=false and Agent's_act=false and abs(Agent's_att - Neighbor's_att) <bounded-
confidence) [
    two-way-communication-process Neighbor's_att Agent's_att]; Scenario #4
end

; Convergence-rate is defined as the decreasing distance between an agent's and neighboring agent's attitudes.
; The agent and its neighboring agent adjust their attitudes based on this opinion distance.
to two-way-communication-process [B C]
    set att round((C+convergence-rate * (B - C))); The attitudes of both the agent and
    set [att] of object round((B+convergence-rate * (C - B))); its neighboring agent change.
end

; The agent's attitude becomes more positive and closer to its neighboring agent's attitude, which
remains the same.
to one-way-persuasion-process-1 [B C]
    set att round((C+convergence-rate * (B - C))); The agent's attitude changes.
end

; The agent's attitude remains the same, and
;       its neighboring agent's attitude becomes more positive and closer to the agent's attitude.
to one-way-persuasion-process-2 [B C]
    set [att] of object round((B+convergence-rate * (C - B))); The neighboring agent's attitude changes.
end

; Phase 4: An agent decides whether or not to adopt an idea or purchase a product.
to adoption-decision-making
    if (act=false) [
      ; when the agent has a positive attitude
      if (att>50) [
        if (count link-neighbors!=0) [
          ; when the proportion of neighboring agents who adopt exceeds the adoption threshold, the
agent also adopts it.
          if ((count link-neighbors with [act=true] / count link-neighbors)>= (theta / 100)) [
            set act true
          ]
        ]
      ]
    ]
end
```

Opinion exchanges between an agent and a randomly chosen neighboring agent can be divided into four scenarios based on adoption status. When one agent has already adopted an idea or product and another has not yet done so, the first is perceived as having a more positive attitude than the second, whose attitude becomes more positive due to a

"user-friendly testimony" effect. The first agent's positive attitude remains the same during and after the opinion exchange phase. Scenarios 1.1 and 2.2 in Algorithm 3 are both consistent with a one-way persuasion process, indicating that only the not-yet-adopted agent's attitude is affected by the opinion exchange phase.

An interesting situation arises when the second agent's attitude is more positive than the first. Following the opinion exchange phase, the not-yet-adopted agent's attitude becomes less positive, and the already-adopted agent's attitude becomes more positive. Scenarios 1.2 and 2.1 both refer to two-way communication, with two key factors involved—the not-yet-adopted agent's adoption threshold is too high, and that same agent has too many neighboring agents, meaning that the proportion of already-adopted neighbors does not exceed the adoption threshold. When both an agent and its neighboring agent have already adopted an idea or purchased a product during the opinion exchange phase, the already-adopted agent with the lower attitude value is affected by the already-adopted agent with the higher attitude value, thus becoming more positive. Scenarios 3.1 and 3.2 in Algorithm 3 are both consistent with this one-way persuasion process. In the last situation, two not-yet-adopted agents affect each other as their attitudes move closer during the opinion exchange phase: the agent with the higher attitude value experiences a decrease, and the agent with a lower value an increase.

As shown in Fig 3 and Algorithm 3, the opinion exchange phase is immediately followed by an adoption decision-making phase, during which a not-yet-adopted agent is sufficiently affected to adopt or purchase an idea or product under two conditions: it has a positive attitude, and the proportion of neighboring agents who have already done so exceeds the agent's adoption threshold.

Note that during each simulation time step, all agents execute communication and adoption decision-making procedures via pseudo-concurrent updating. The order of agent processing is randomly shuffled during each tick, with modifications of each agent's state immediately taking effect. This random shuffling process was added using the *ask-concurrent* command in NetLogo 4.0.5; the behavior is faithfully reproduced by Python 3. This scheme is positioned between strictly synchronous updating (with all agents reading old states prior to writing) and strictly sequential updating according to a fixed order (with queue position systematically providing advantages to certain agents).

This random reshuffling of an execution order-per-tick scheme was designed to ensure that no agent systematically benefits from its processing sequence position. Simulation randomness is thus based on four well-defined stochastic elements: the random selection of a neighbor for every agent during each tick, the initial random assignment of attitude and threshold values, the random or clustered placement of pioneer agents, and random execution order during each tick. Since each source is controlled by and reproduced according to a fixed random seed, the sensitivity analysis data reported in the Results section reflect the effects of model parameters rather than systematic scheduling.

## Evaluation indicators

After adding specific input parameters, interaction rules, and an update scheme, two evaluation indicators are defined to facilitate model simulation process assessment and sensitivity analyses: favorable reviews (FRI) and good sales (GSI). By respectively quantifying agent attitudes and adoption behaviors, these indicators provide a comprehensive framework for analyzing the potential success of an innovative product or idea. FRI quantifies agent proportions in populations showing positive attitudes toward the product or idea being examined, calculated as the percentage of agents whose attitude values ($att$) exceed 50, and expressed as

$$\text{FRI} = \frac{\sum_{i=1}^{N} \mathbb{I}(\text{att}_i > 50)}{N}$$

(3)

where $N$ is the total number of agents in the agent population, $att_i$ the attitude value of the $i$-th agent ($i = 1, 2, \ldots, N$), and $\mathbb{I}(\bullet)$ a binary function equal to 1 when $att_i > 50$ and 0 otherwise. The result is a quantitative representation of an agent population's sentiment toward a product or idea—the higher the value, the stronger the sentiment and the more favorable the opinion.

GSI values indicate the proportions of agents in a population who have already adopted an innovative product or idea, calculated as the percentage of agents whose adoption status (*act*) is True. It is mathematically expressed as

$$\text{GSI} = \frac{\sum_{i=1}^{N} \mathbb{I}(\text{act}_i = \text{True})}{N} \tag{4}$$

where $\text{act}_i$ denotes the adoption status of the *i*-th agent and $\mathbb{I}(\bullet)$ a binary function equal to 1 if $\text{act}_i$ = True, 0 otherwise. This indicator of a product or idea's actual degree of adoption reflects the effectiveness of the diffusion process being examined, with a higher GSI value indicating greater success in convincing agents to take action.

As shown in Algorithm 3, the relationship between FRI and GSI is based on an agent's adoption decision-making logic. Specifically, during a simulation run the agent's adoption status is contingent on achieving a positive *att* (>50) toward the product or idea, plus the adoption statuses of its neighboring agents. This requirement ensures that all agents who adopt the product in question represent a subset of all agents holding favorable attitudes, meaning that the proportion of agents with positive attitude values will always be greater than or equal to the proportion of adopters—that is, $0 \le \text{GSI} \le \text{FRI} \le 1$. Accordingly, it is sufficient to only use the easily observable and illustratable GSI to evaluate sales strength when assessing BCAT model simulation processes and outcomes.

Another quantifiable simulation trajectory characterizing collective adoption outcomes is critical point ($t^*$), defined as the first time step where GSI exceeds 0.5—in other words, the first *t* at which adopters constitute a clear agent population majority. This threshold value is computed and displayed in real time by the simulation interface, and recorded as part of the output of each run. $t^*$ marks the transition from adoption as a minority behavior to adoption as a mainstream social norm, in the sense of Rogers' [7] diffusion stages. If this value is not achieved, then $t^*$ is recorded as zero. This is considered the defining signature of "best game no one played" outcomes, with the adoption process failing to achieve majority diffusion regardless of how favorable the opinion climate might be.

## Emergent properties of the combined model

The BCAT model has two properties that are not individually present in either component model, the first being a dual-filter mechanism along the attitude formation-to-adoption path. The bounded confidence component serves as a first-stage filter that partitions populations into opinion clusters. Agents whose attitudes stabilize below the positive threshold ($att \le 50$) become permanently ineligible for adoption, thus shrinking the pool of potential adopters in the FRI subset. The adoption threshold component serves as a second-stage filter operating within this subset: even agents with positive attitudes will fail to adopt the idea or product if the proportions of already adopting neighbors do not exceed their individual thresholds. This sequential filtering supports the above-established inequality $0 \le \text{GSI} \le \text{FRI} \le 1$. It also provides a mechanistic interpretation of the inequality: adoption failure can arise from opinion fragmentation, coordination failure, or both mechanisms acting simultaneously. Neither a pure opinion dynamics model nor a pure adoption threshold model is capable of generating a compound relationship consisting of opinion and adoption outcomes.

The second emergent property is an adoption-to-opinion feedback loop that is not found in either component model. As shown in Algorithm 3, Scenarios 1.1 and 2.2 exhibit a user-friendly testimony effect in which an already-adopted agent exerts a one-way positive influence on the attitude of a not-yet-adopted neighbor, but only when their attitude distance falls within the bounded confidence threshold. Accordingly, adoption events selectively propagate the influence of a positive attitude, accelerating agent convergence within the adopter's opinion neighborhood, but failing to reach agents in distant opinion clusters. The result is mutual dependence between the two sub-processes: opinion dynamics controls which agents are eligible for and receptive to adoption, and which adoption events reshape opinion trajectories among topologically and attitudinally reachable neighbors. Constrained by the bounded confidence threshold, this bidirectional coupling generates path-dependent and stochastically sensitive adoption dynamics, with identical initial parameter distributions

capable of yielding qualitatively divergent outcomes depending on the spatial arrangement of early adopters relative to opinion cluster boundaries.

## Results

In this section we present BCAT model simulation results for two contrasting scenarios: favorable reviews with good sales and favorable reviews with poor sales. The purpose is to demonstrate how the dual-filter mechanism and feedback loop described above manifest in simulation trajectories.

### Simulating favorable reviews with good or poor sales

Fig 4 shows a regular lattice simulation using parameters designed to establish win-win conditions for high approval and high adoption—in other words, the synergistic effect of a specific parameter configuration. As shown, agent attitudes

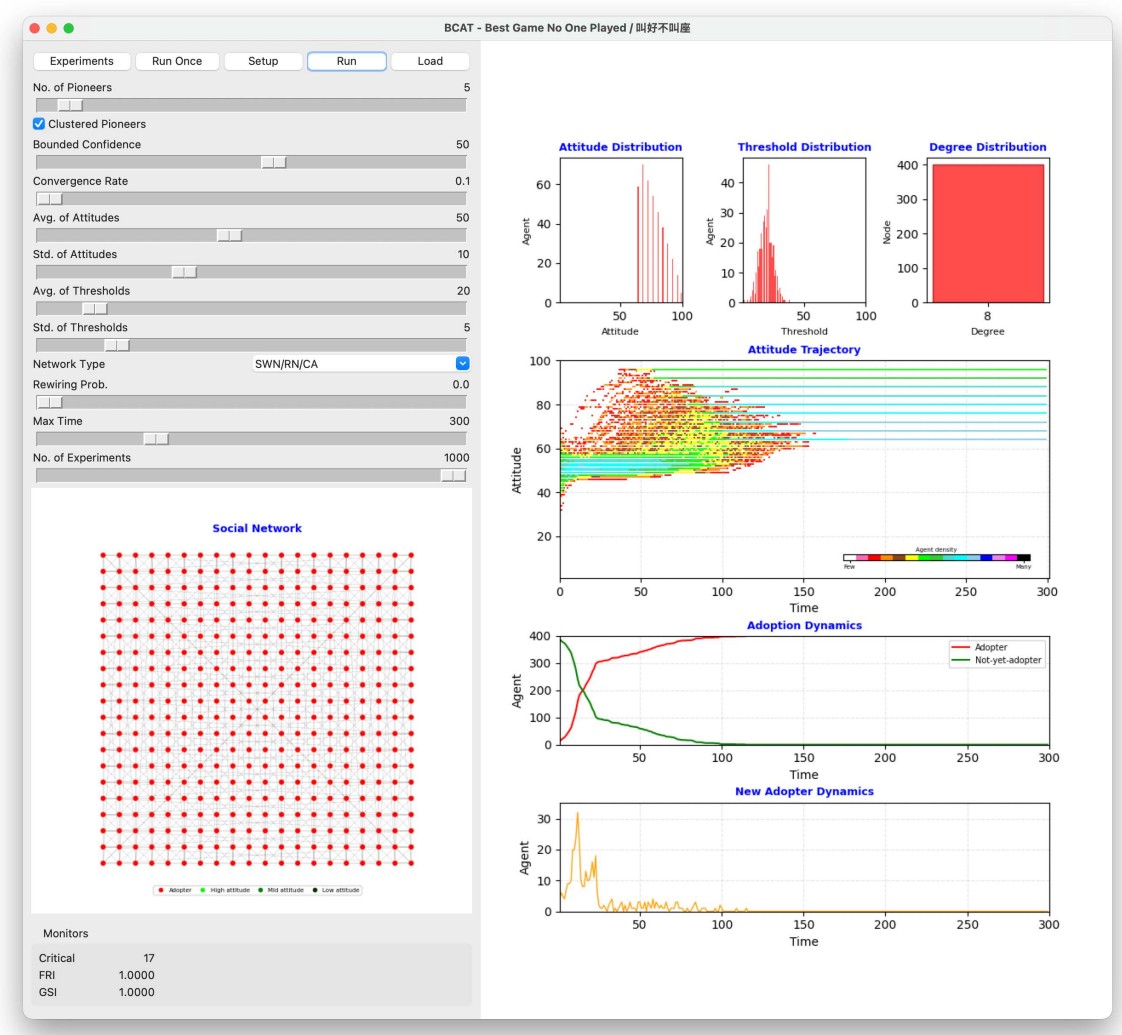

**Fig 4. Simulation result illustrating favorable review and good sales scenario.**

converged toward higher values over time, indicating widespread product approval. The attitude trajectory graph shows a steady increase in average attitude from an initial mean of 50. Adoption dynamics mirror this trend—in this particular example a critical number of 17 adopting agents emerged early in the simulation, thus triggering a cross-network cascade effect. This result aligns with a critical mass theory scenario in which a small number of adopters are capable of generating widespread acceptance under favorable conditions. All agents adopted the product by the end of the simulation (i.e., GSI = 1), an example of a complete in-network adoption diffusion process.

The simulation shown in Fig 4 underscores the importance of parameter interaction in shaping opinion dynamics and adoption behaviors. The relatively high bounded confidence value facilitated extensive opinion exchanges, thus reducing polarization while encouraging collective agreement. The moderate average attitude value ensured a balanced starting point, thus avoiding initial biases that might have skewed the results. Similarly, the low average adoption threshold resulted in early adopters exerting significant influence and creating the necessary conditions for the observed adoption cascade. The regular lattice structure contributed to this outcome by ensuring uniform agent interaction, thereby promoting equitable cross-network levels of attitude and adoption behavior diffusion.

The scenario in Fig 5 shows positive alignment in agent attitudes toward a product, but a minimal adoption end result. Simulation initialization began with agent attitude and adoption threshold values drawn from normal distributions, thus shaping each agent's predisposition to product adoption and susceptibility to social influences. The bounded confidence parameter allowed agents with attitude differences below 50 to engage in opinion exchanges, resulting in gradual attitude convergence. During the simulation, average agent attitude increased from an initial value of 50, indicating a collective movement toward product approval. However, adoption dynamics data revealed a starkly different picture: only 11 of 400 agents adopted the product by the end of the simulation—a meager 2.75% adoption rate (GSI = 0.0275).

Adoption failure following a positive attitude trajectory can be attributed to a high average adoption threshold. At a mean threshold of 40%, agents required at least 40% of their neighbors to adopt a product before making their own adoption decisions. The combination of this threshold and a regular lattice structure produced a significant adoption diffusion barrier. The uniform connectivity structure of the lattice ensured consistent connections among agents, but also limited the ability of early adopters to exert their influence beyond their immediate neighborhoods.

This reflects the absence of a critical mass of early adopters, a key factor in triggering widespread diffusion. Critical mass theory suggests that once a sufficient number of agents adopt a product or idea, their collective influence has greater potential for creating a cross-network adoption cascade. No such critical point emerged in this particular simulation, therefore early adopters remained isolated from their neighbors. This outcome underscores the importance of adoption thresholds and network topologies in determining diffusion process success. While the bounded confidence parameter promoted attitude convergence, it was insufficient for overcoming the network's high adoption thresholds and structural constraints.

These findings highlight the challenges of translating positive attitudes into widespread adoption. Favorable opinions alone are insufficient for adoption diffusion success in the presence of structural and behavioral barriers such as high adoption thresholds and uniform network connectivity. This particular case emphasizes the need for strategies that lower adoption thresholds and/or introduce network randomness to enhance innovation diffusion.

## Two simulations with identical initial conditions

Fig 6 shows two simulations with identical parameter settings that were executed to demonstrate BCAT model sensitivity to initial conditions. Despite condition consistency, the two simulations yielded remarkably different results, one with a favorable good sales outcome, one with a best game no one played outcome. As shown in Fig 6a, agent attitudes converged towards higher values as the simulation progressed. The adoption dynamics plot reveals a clear critical point at which the presence of 46 adopters triggered a cascade of adoption behaviors throughout the network. Consequently, all

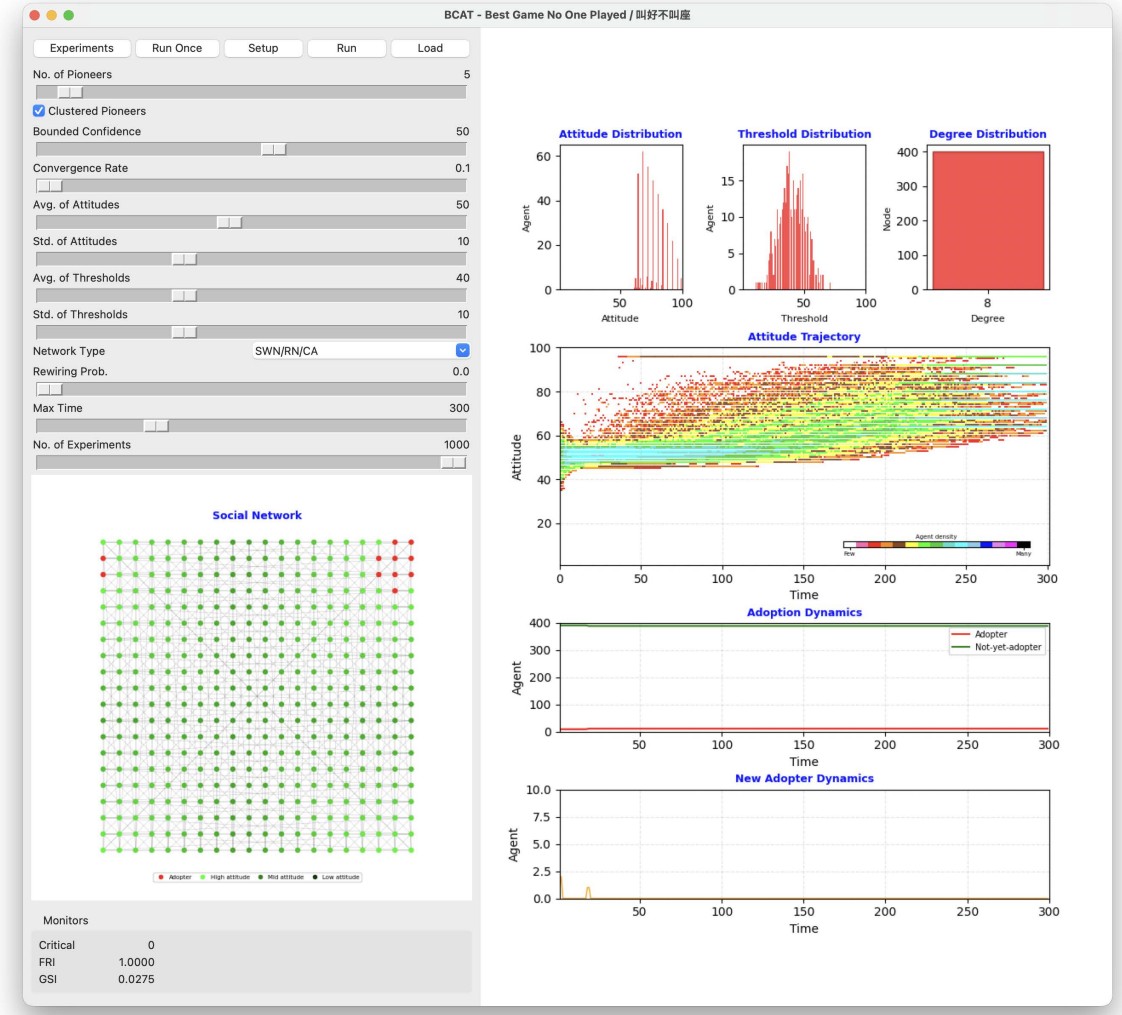

**Fig 5. Simulation result illustrating favorable review but poor sales scenario.**

agents adopted the product by the end of the simulation (i.e., GSI = 1), another example of positive attitudes producing high adoption rates.

Fig 6b shows contrasting outcomes despite identical parameter settings. Although attitudes still converged towards higher values, adoption dynamics were fundamentally different, with the absence of a critical point indicating that early adopters failed to generate the necessary momentum for widespread diffusion. The simulation ended with only a small percentage of agents adopting the product, despite a favorable average attitude. The divergence may be attributed to subtle differences in the initial agent attitude and adoption threshold distributions. Though small, these differences resulted in isolated clusters of adopters that were incapable of propagating network-wide adoption behaviors among their neighbors (GSI = 0.0125).

In summary, the identical parameter settings were conducive to widespread adoption in theory, but the actual outcomes were shaped by the stochastic nature of agent initialization. These findings reflect BCAT model sensitivity to primary

(a)

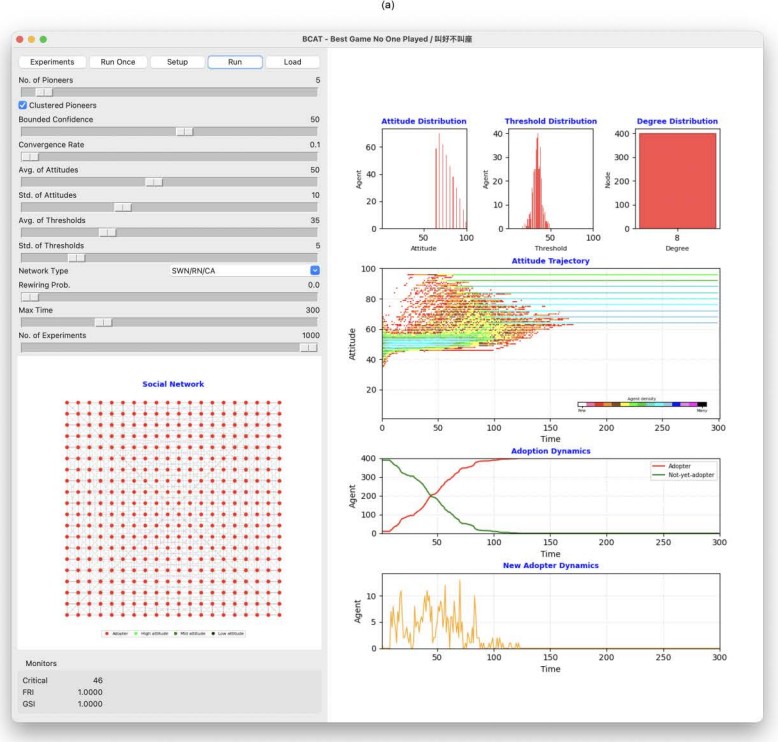

(b)

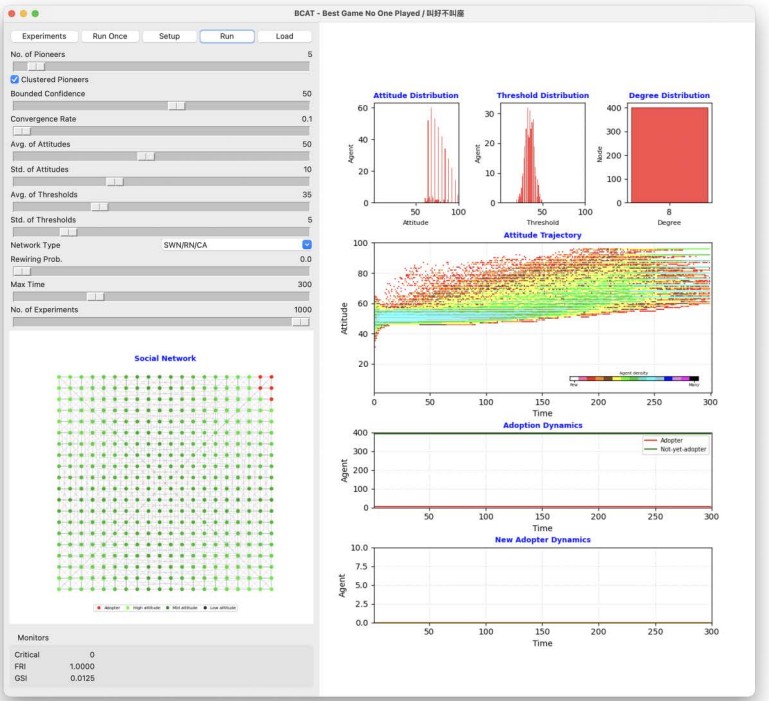

**Fig 6. Two simulation runs with identical parameter settings but different processes and results: (a) favorable review and good sales scenario, (b) favorable review but poor sales scenario.**

parameters such as attitude and adoption threshold, even within controlled network structures. They also emphasize the dual importance of network topology and critical mass in promoting collective behavior. The two simulation results demonstrate how different combinations of model-related parameters, network structure, and initial conditions drive adoption dynamics. Outcome divergence reveals the challenges of achieving both widespread approval and adoption, even under identical conditions.

## Sensitivity analysis

Table 3 presents results from our examination of the roles of five BCAT model parameters (bounded-confidence, avg-of-attitudes, std-of-attitudes, avg-of-thresholds, std-of-thresholds) in shaping adoption behaviors in regular lattice, small-world, random, and scale-free networks. Five complementary statistical methods were applied: feature importance from a Random Forest regression ensemble (100 trees, mean decrease in impurity criterion, default hyperparameters),

**Table 3. Results from statistical analysis of primary model-related parameters.** For Feature Importance entries, the standard error of per-tree importance estimates is ≤0.001 for all entries (100-tree Random Forest ensemble, $N > 15,000$ per network topology); the remaining four methods produce deterministic analytical results, so standard errors are not applicable. Parameter Importance values (Pearson correlations) closely approximate Standardized Regression coefficients because the five input parameters are varied independently in the sensitivity analysis design, making them approximately uncorrelated; under orthogonal predictors, the two measures are mathematically equivalent. All *avg-of-thresholds* values are stored as integers in [1, 100] and are equivalent to fractional adoption thresholds in [0.01, 1.00] via the conversion $\theta = avg - of - thresholds / 100$, as implemented in the adoption decision rule of Algorithm 3; the sensitivity analysis sweeps *avg-of-thresholds* across the range [10, 70], corresponding to fractional thresholds of [0.10, 0.70].

| Analysis Method | Network Type | bounded-confidence | avg-of-attitudes | std-of-attitudes | avg-of-thresholds | std-of-thresholds |
|---|---|---|---|---|---|---|
| Feature Importance | **All** | **0.17** | **0.20** | **0.04** | **0.54** | **0.05** |
| | Regular lattice | 0.20 | 0.20 | 0.05 | 0.55 | 0.01 |
| | Small-world | 0.19 | 0.19 | 0.05 | 0.56 | 0.01 |
| | Random | 0.16 | 0.16 | 0.04 | 0.53 | 0.11 |
| | Scale-free | 0.13 | 0.24 | 0.04 | 0.51 | 0.08 |
| Multivariate Regression | **All** | **0.37** | **0.53** | **0.21** | **−1.26** | **0.47** |
| | Regular lattice | 0.43 | 0.51 | 0.16 | −1.35 | −0.05 |
| | Small-world | 0.41 | 0.51 | 0.17 | −1.35 | 0.15 |
| | Random | 0.33 | 0.47 | 0.17 | −1.26 | 0.73 |
| | Scale-free | 0.33 | 0.56 | 0.25 | −1.21 | 0.64 |
| Partial Correlation | **All** | **0.31** | **0.38** | **0.07** | **−0.70** | **0.15** |
| | Regular lattice | 0.39 | 0.36 | 0.05 | −0.71 | −0.02 |
| | Small-world | 0.38 | 0.37 | 0.06 | −0.73 | 0.05 |
| | Random | 0.32 | 0.34 | 0.06 | −0.70 | 0.24 |
| | Scale-free | 0.25 | 0.41 | 0.08 | −0.69 | 0.21 |
| Standardized Regression | **All** | **0.22** | **0.27** | **0.05** | **−0.65** | **0.10** |
| | Regular lattice | 0.27 | 0.25 | 0.03 | −0.66 | −0.01 |
| | Small-world | 0.26 | 0.25 | 0.04 | −0.67 | 0.03 |
| | Random | 0.22 | 0.24 | 0.04 | −0.65 | 0.16 |
| | Scale-free | 0.17 | 0.30 | 0.06 | −0.64 | 0.14 |
| Parameter Importance | **All** | **0.22** | **0.27** | **0.05** | **−0.65** | **0.10** |
| | Regular lattice | 0.27 | 0.25 | 0.03 | −0.66 | −0.01 |
| | Small-world | 0.26 | 0.25 | 0.04 | −0.67 | 0.03 |
| | Random | 0.22 | 0.24 | 0.04 | −0.65 | 0.16 |
| | Scale-free | 0.17 | 0.30 | 0.06 | −0.64 | 0.14 |

ordinary least squares multivariate regression coefficients, partial correlations controlling for co-variation among pre-
dictors, standardized regression coefficients (beta weights), and bivariate Pearson correlation coefficients (parameter
importance). The data represent analyses of each parameter's importance, sensitivity, and network-dependent influence.
As shown in Table 3, average adoption threshold consistently proved to be the primary determinant of adoption behav-
iors across all network structures. As an indicator of mean level of within-network conservatism, this parameter is a direct
determinant of required thresholds for agent adoption of innovative products or ideas. According to the detailed correla-
tion matrix and analysis results for all four network types shown in Figs 7 and 8, there is a strong negative relationship
between avg-of-thresholds and adoption outcome (in this case, sales), underscoring its inhibitory role. As shown in Fig 9,

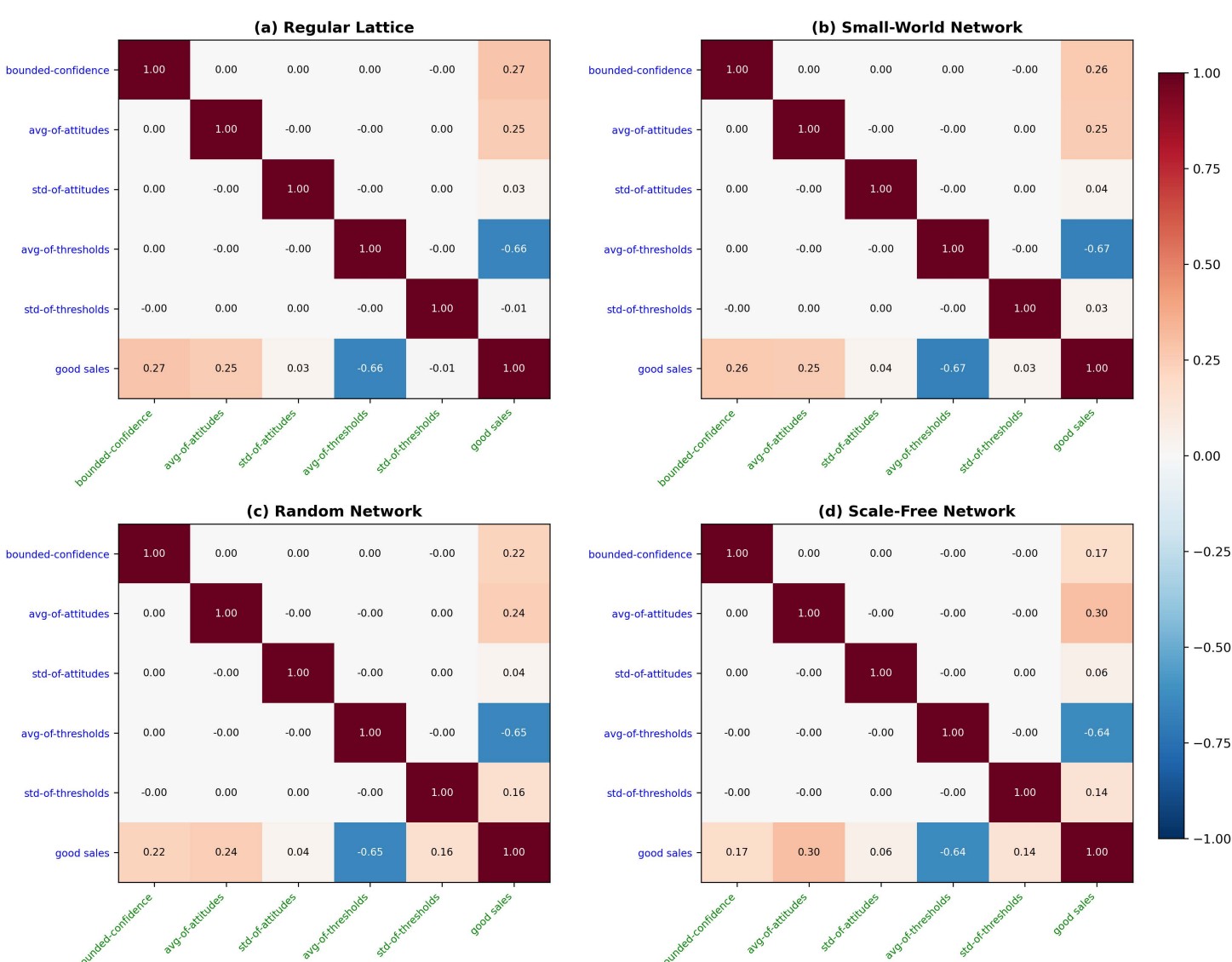

**Fig 7. Correlation coefficient heatmaps showing various primary model-related parameters and good sales indicator (GSI) values across four
network types: (a) regular lattice, (b) small-world, (c) random and (d) scale-free.**

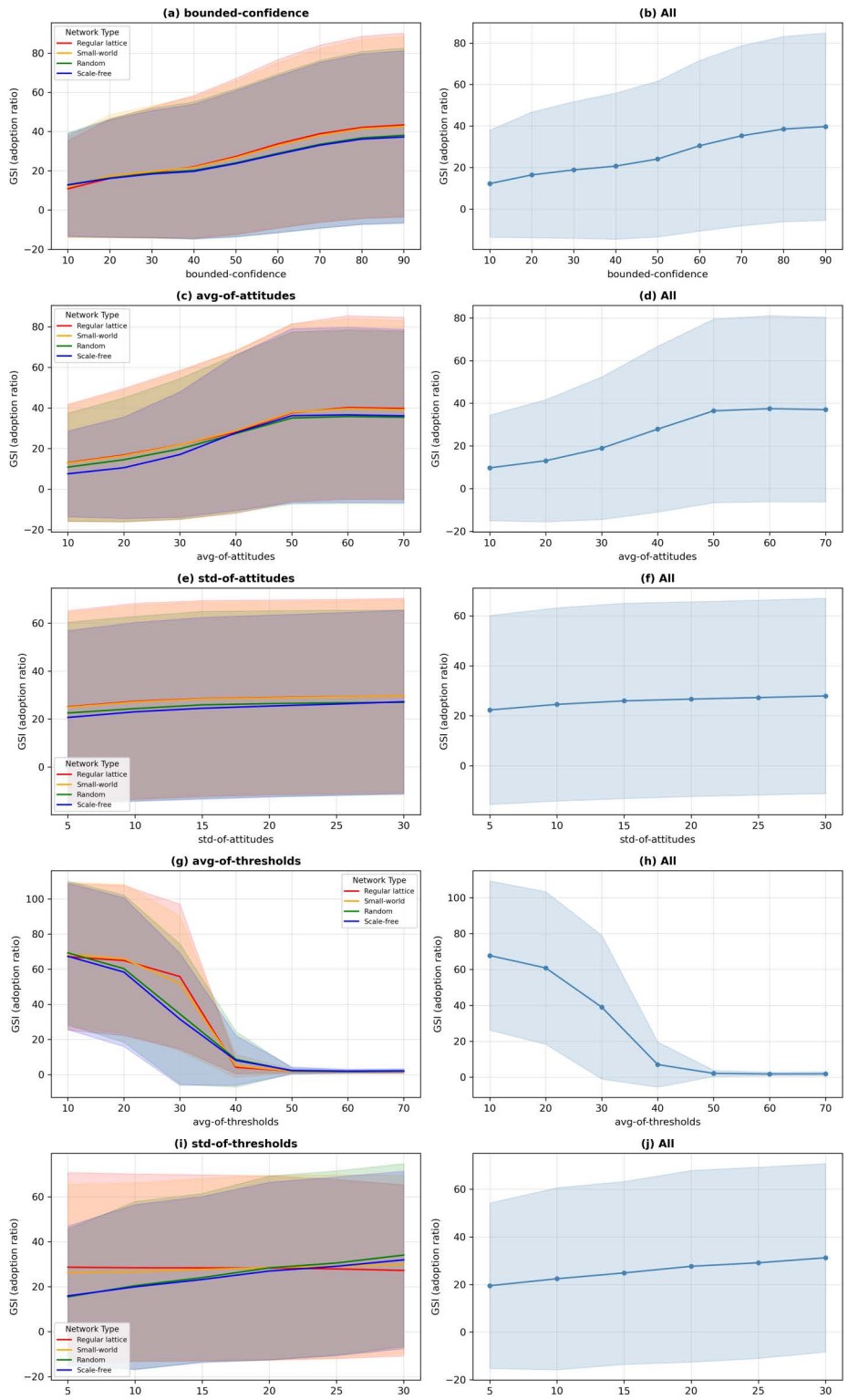

**Fig 8. Sensitivity analysis results for various primary model-related parameters and good sales indicator (GSI) values across four network types (regular lattice, small-world, random and scale-free).** (a, b) bounded-confidence, (c, d) avg-of-attitudes, (e, f) std-of-attitudes, (g, h) avg-of-thresholds, (i, j) std-of-thresholds. Left plots show results grouped by network type; right plots show aggregated results across all networks.

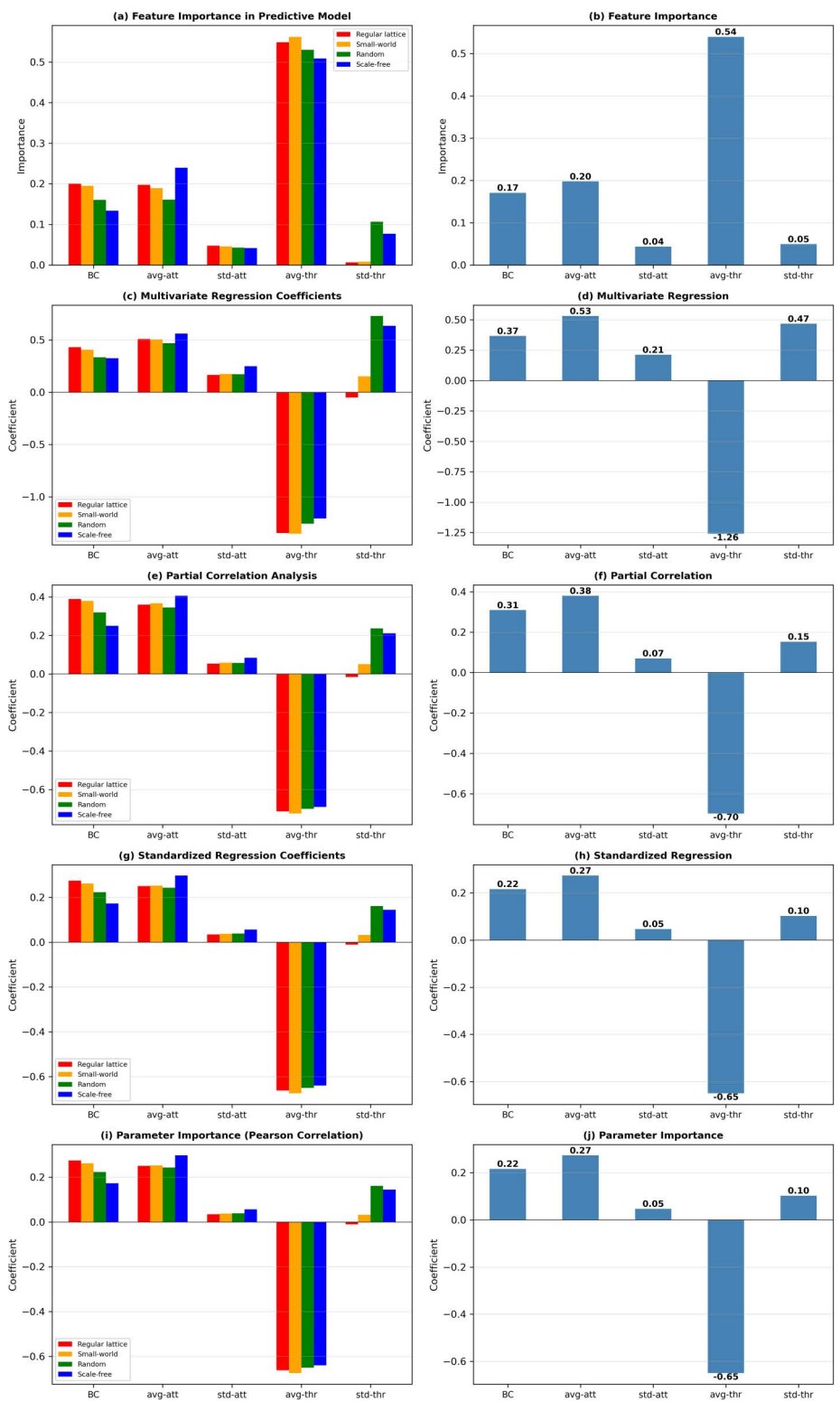

**Fig 9. Statistical analysis results for primary BCAT model parameters.** (a, b) feature importance, (c, d) multivariate regression, (e, f) partial correlation, (g, h) standardized regression, (i, j) parameter importance. Left plots: experiment data are grouped according to network type for statistical analysis. Red bar represents regular lattice, yellow bar small-world, green bar random, and blue bar scale-free networks. Right plots: statistical analysis results for entire body of experimental data.

this negative effect was further identified by multivariate and standardized regression coefficient data, with avg-of-thresholds consistently exhibiting the strongest effect of all examined parameters.

The feature importance analysis results shown in Fig 9 indicate a pronounced dominance of avg-of-thresholds in regular lattice and small-world networks, where localized interactions and constrained information flow amplified the effects of higher thresholds on adoption suppression. Those same results also indicate that avg-of-thresholds was the most influential parameter in random and scale-free networks, but its impact was mitigated by the enhanced connectivity. These findings confirm the universal importance of this parameter in governing the speed and extent of innovation diffusion regardless of network topology.

Average attitude plays a pivotal complementary role in influencing adoption behaviors, reflecting the initial predispositions of agents toward innovation while serving as a baseline for subsequent opinion dynamics. Positive correlations between avg-of-attitudes and adoption outcomes were observed across all networks, suggesting that higher average attitudes promote more favorable adoption environments (Figs 7 and 8). This effect was particularly strong in scale-free networks (Fig 7d), where hub nodes amplified favorable initial attitudes and catalyzed rapid innovation diffusion.

The regression coefficient data shown in Fig 9 underscore the consistent yet moderate impact of avg-of-attitudes. In regular lattice and small-world networks characterized by localized opinion exchanges, the influence of avg-of-attitudes is attenuated, but its role in aligning initial opinions with adoption tendencies remains essential. By establishing favorable initial conditions, this parameter ensures a smoother opinion alignment trajectory that facilitates subsequent adoption behaviors. Also as shown in Fig 9, the standard deviation of adoption thresholds parameter, which reflects diversity in agent conservatism, exhibited significant sensitivity to structural network characteristics. In random and scale-free networks marked by extensive and efficient information flow, std-of-thresholds exhibited a positive correlation with adoption outcome, suggesting that greater threshold diversity encourages dynamic and widespread adoption patterns. In contrast, in regular lattice and small-world networks characterized by limited connectivity, threshold diversity failed to effectively encourage dynamic and widespread adoption patterns, resulting in minimal impact on adoption outcomes. An analysis of feature importance corroborates these findings—that is, the significance of std-of-thresholds was amplified in high-connectivity networks (Fig 9). The role of this parameter as a sensitivity factor emphasizes its potential for driving heterogeneous adoption behaviors in well-connected systems, while its limited effects in localized networks reflect topologically-imposed constraints.

The bounded-confidence parameter, reflecting agent tolerance for opinion differences, showed consistent positive correlations with adoption outcomes across all network structures (Figs 7 and 8). Bounded-confidence supports interaction dynamics considered essential for adoption decisions by enabling agents to engage in meaningful opinion exchanges. The multivariate regression results shown in Figs 9c and 9d confirm its moderate but stable influence, with slight variation observed across different network types. Its impact was lower in scale-free networks, where hub nodes dominate information flow, and where opinion exchanges become less critical for adoption (Figs 7d and 8a). In regular lattice (Figs 7a and 8a) and small-world networks (Figs 7b and 8a), bounded-confidence assumed greater significance due to its reliance on localized interactions. As shown in Fig 9, these observations underscored the dual roles of this parameter as a stable facilitator of opinion exchanges, as well as a context-dependent factor influenced by network topology.

The standard deviation of attitudes parameter, an indicator of initial opinion heterogeneity, consistently demonstrated a limited effect on adoption outcomes (Figs 7, 8e and f). The correlation and regression analysis data shown in Figs 7–9 show weak associations between std-of-attitudes and adoption results. This marginal impact may be attributable to the dynamic nature of opinion exchanges in the BCAT model, which quickly overrides initial heterogeneity with threshold-driven activity. However, in localized networks (regular lattice, small-world), std-of-attitudes makes a contribution to diversity in local interactions (Fig 8e), thereby adding a layer of complexity to adoption dynamics. Although its overall influence is minor, its role in shaping early opinion exchanges deserves attention in specific contexts.

                                        

The interplay between parameter effects and network topology is of central importance in the BCAT model. As shown in Fig 9b, parameters such as avg-of-thresholds performed uniformly dominant roles across all networks, highlighting their universal importance. In contrast, the influence of std-of-thresholds (Figs 8i and 9i) and bounded-confidence (Figs 8a and 9i) varied significantly depending on network connectivity. High-connectivity networks (random and scale-free) amplified the effects of threshold diversity, resulting in broader adoption. Weaker connectivity in regular lattice and small-world networks limited their impacts, reflecting the primacy of average thresholds and localized interactions.

As shown in Table 3 and Fig 9, BCAT model parameters exhibited distinct roles, sensitivities, and degrees of influence as shaped by network topology. Avg-of-thresholds stands out as the most important determinant of adoption dynamics, with avg-of-attitudes providing essential support in the form of initial opinion alignment. Std-of-thresholds demonstrated high sensitivity to connectivity, with its significance amplified in high-connectivity networks. Although bounded confidence offered a stable interaction foundation, it was influenced by structural constraints. Std-of-attitudes played a minor but contextually relevant role. These findings illustrate the intricate interplay of parameters with network structures, providing valuable information for optimizing the BCAT model for further exploration of diffusion.

To verify the robustness of these findings at various system sizes, we conducted scaling experiments with $N = 900$ ($30 \times 30$), $N = 1,600$ ($40 \times 40$), and $N = 2,500$ ($50 \times 50$) agent populations. Results confirmed the above-described qualitative phenomena, including the opinion–adoption gap, the phase transition at intermediate adoption thresholds, and the dominance of *avg-of-thresholds* in sensitivity analyses. All were found to be robust for different system sizes, and to not be artifacts of the $N = 400$ baseline configuration. Scaling data are available in a public repository (Doi:10.5281/zenodo.19216365).

## Mechanism decomposition: Coordination failure versus opinion clustering

While sensitivity analysis results demonstrate the dominance of avg-of-thresholds as an adoption outcome determinant, they did not distinguish between the contributions of the two mechanisms through which BCAT generates opinion–adoption gaps. To separate these channels we designed three controlled experiments (MD-A, MD-B, MD-C) for selectively activating or suppressing each mechanism in regular lattice and small-world topologies. Each configuration consisted of 1,000 replications and 300 simulation ticks per run.

For the MD-A experiment we isolated a coordination failure channel by initializing all non-pioneer agents with a uniform attitude of 80 (avg-of-attitudes = 80, std-of-attitudes = 0) and setting the bounded-confidence value to 10. The attitude distance between pioneers (*att* = 100) and non-pioneers (*att* = 80) was set to 20, which exceeded the communication threshold so that no opinion exchanges occurred. Since all non-pioneers shared identical attitudes, peer communication among them also failed to produce changes. According to this configuration, an FRI value of 1.0 was guaranteed throughout all simulations, and any adoption failure could be attributed to threshold-based coordination failure. We swept through avg-of-thresholds values from 10 to 70 in increments of 10, with std-of-thresholds = 10 and no-of-pioneers = 5 (clustered).

For the MD-B experiment we isolated the opinion clustering channel by setting no-of-pioneers to 0, thus eliminating all adoption seeds. In the complete absence of agent adopting activity, the adoption decision condition was never satisfied and GSI remained zero throughout. Opinion dynamics therefore operated in complete isolation from adoption feedback. Parameter values were set at avg-of-attitudes = 50, std-of-attitudes = 15, and bounded-confidence = 50, consistent with the main sensitivity analysis.

For the MD-C experiment we ran the full BCAT model with the same opinion and network parameters as in the MD-B experiment, but added no-of-pioneers = 5 (clustered) and swept through avg-of-thresholds values from 10 to 70. The act of measuring both FRI and GSI supported a within-model decomposition in which the opinion clustering contribution = $1 - \text{FRI}$, and the coordination failure contribution = $\text{FRI} - \text{GSI}$.

The MD-A experiment results for the regular lattice network revealed a sharp phase transition in coordination dynamics between avg-of-thresholds values of 30 (GSI = 0.976) and 40 (GSI = 0.026) (Fig 10a). At low thresholds (10–20) the

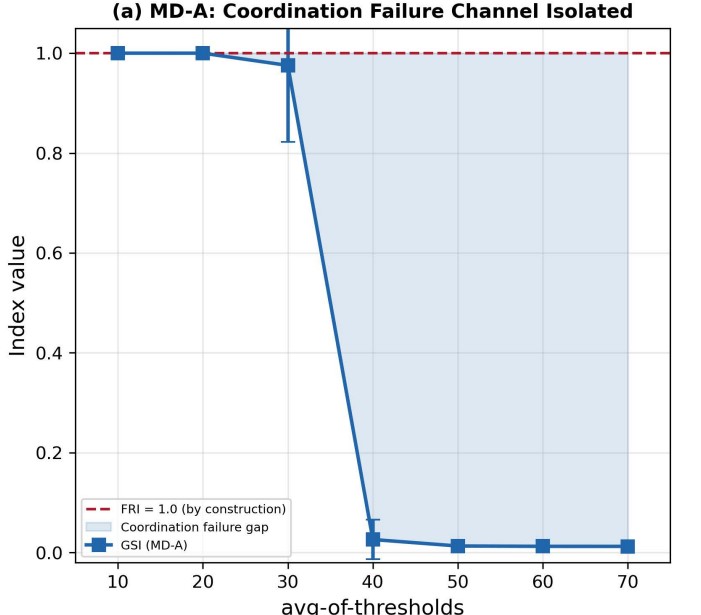
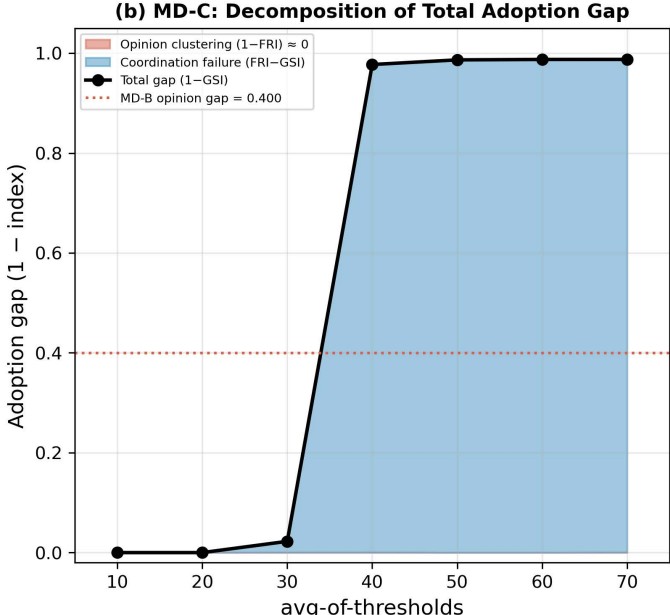

**Fig 10. Mechanism decomposition on the regular lattice.** (a) MD-A isolates the coordination failure channel: FRI = 1.0 by construction, and GSI declines sharply between avg-of-thresholds = 30 (GSI = 0.976) and 40 (GSI = 0.026), indicating a phase transition in coordination dynamics. (b) MD-C decomposes the total adoption gap (1 − GSI) into opinion clustering (1 − FRI, red) and coordination failure (FRI − GSI, blue). The opinion clustering contribution is negligible because the user-friendly testimony effect (Algorithm 3, Scenarios 1.1 and 2.2) restores FRI to approximately 1.0 even though MD-B alone yields FRI = 0.600 (dotted line). N = 400 agents, 1,000 runs per point.

adoption cascade was completely propagated (GSI = 1.0). At high thresholds (40–70) only the five pioneer agents adopted the item in question (GSI ≈ 0.013), indicating total coordination failure.

The MD-B results reveal that opinion clustering alone reduced the eligible adopter pool to FRI = 0.600 in the regular lattice and FRI = 0.618 in the small-world network, confirming the ability of the bounded confidence mechanism alone to reduce the potential adopter population by approximately 40%.

The MD-C results yielded a striking finding for the full BCAT model: FRI values recovered to approximately 1.0 across all threshold values (Fig 10b). This recovery can be explained by the adopting pioneers exerting a user-friendly testimony effect on bounded confidence-reachable neighbors, thus increasing their attitudes above the eligibility threshold. As a result, the opinion clustering contribution (1 − FRI) to the total adoption gap was negligible, and the gap was dominated by a coordination failure channel (FRI − GSI). We observed qualitatively similar decomposition patterns for the small-world network (rewiring probability = 0.10), with the phase transition occurring more gradually at an avg-of-thresholds value of 30 (GSI = 0.825 for a small-world network versus 0.976 for a regular lattice network).

According to the observed decomposition, the testimony effect acts as a built-in corrective mechanism that counteracts opinion clustering when adoption seeds are present. However, this corrective effect is dependent on the quality of the adoption cascade: when coordination failure prevents the cascade from propagating (high thresholds), the testimony effect cannot reach distant agents, and opinion clustering may persist. This conditional interdependence between the two channels (in which adoption feedback can repair opinion fragmentation, but only when coordination succeeds) is a property of the coupled BCAT system—that is, neither component model by itself is capable of this interdependence.

## Downward compatibility with opinion dynamics and adoption threshold models

The BCAT model uses precisely defined parameter configurations to achieve downward compatibility with both a bounded confidence-based opinion dynamics model and an adoption threshold model of innovation diffusion. This feature supports context-dependent BCAT simplification to corresponding versions of the two foundational models, while retaining the ability to replicate their core dynamics. Fig 11 shows results from our effort to make the BCAT model downward-compatible with an opinion dynamics model, focusing exclusively on public opinion evolution. Adoption-related parameters were initially established to ensure that no adoption behaviors occurred during the simulation—specifically,

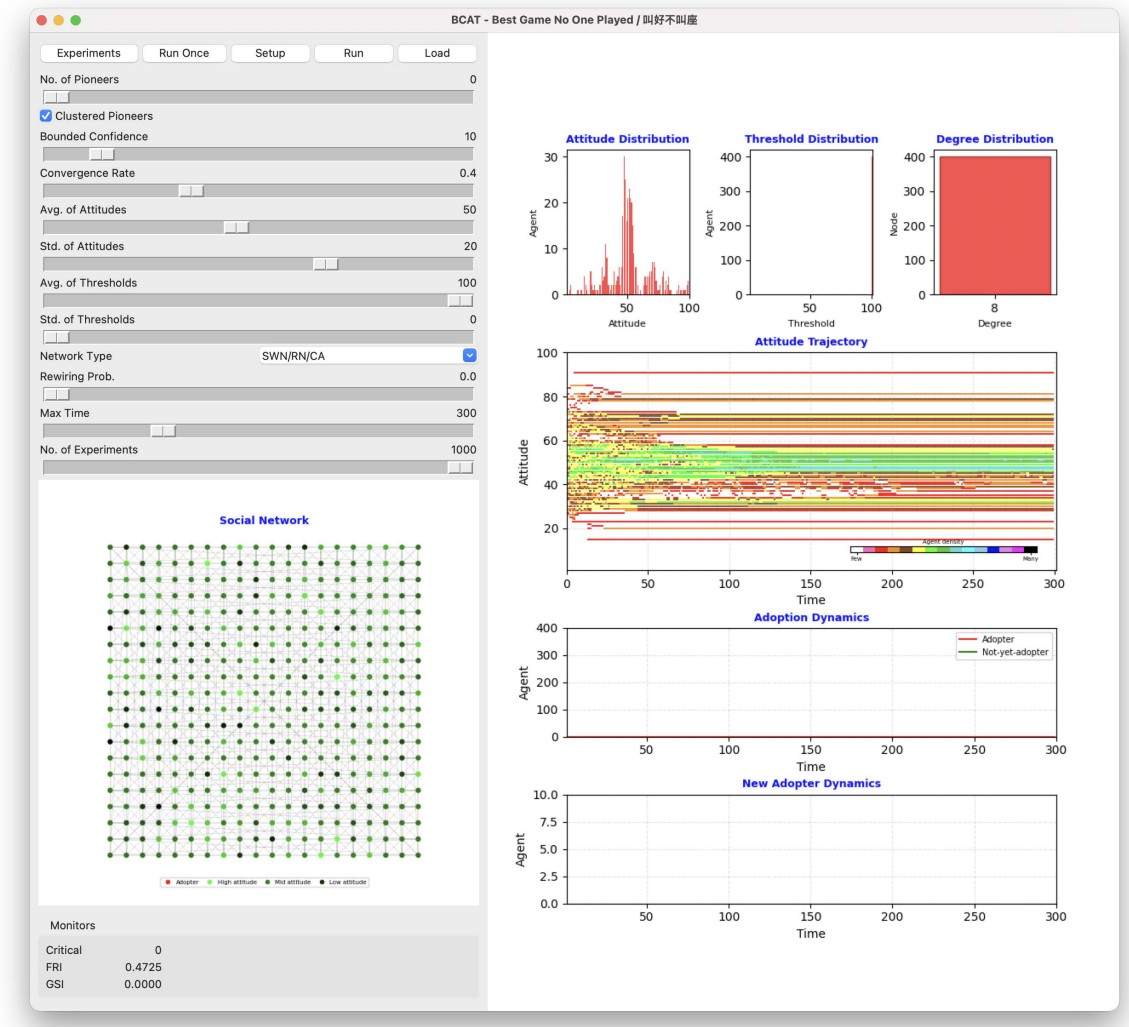

**Fig 11. BCAT model configured as a bounded confidence-based opinion dynamics model (adoption dynamics removed).** Adoption thresholds are unattainable (avg-of-thresholds = 100, std-of-thresholds = 0) and initial adopters are absent (no-of-pioneers = 0), ensuring a complete model focus on opinion dynamics. Opinions evolve under the influence of a bounded confidence mechanism (bounded-confidence = 10) with a convergence rate of 0.4. The network is a regular lattice generated with a rewiring probability of 0.00. Each of the 400 agents is connected to an average of 8 neighbors. As shown in attitudes trajectory and distribution data, agent attitudes were clustered over 300 time ticks, thus highlighting the bounded confidence effect on opinion clustering in a structured network.

avg-of-thresholds was set to 100 and std-of-thresholds to 0, thus setting all agent adoption thresholds to the extremely high value of 100. Number of initial adopters (no-of-pioneers) was set to 0, again with the result of eliminating any possibility of adoption processes so that the sole simulation focus was on opinion dynamics. Among opinion-related parameters, bounded-confidence (the tolerance range for agents to engage in opinion exchanges) was set to 10, and convergence rate (the rate at which agents adjusted their attitudes after each exchange) to 0.4. Initial agent attitudes followed a normal distribution (avg-of-attitudes set to 50 and std-of-attitudes to 20), indicating a highly diverse initial opinion distribution. As the simulation progressed, opinion exchanges and attitude adjustments occurred at the above-defined rates, resulting in a gradual contraction in attitude distribution over time.

Our results show that the standard deviation of attitudes decreased steadily from the initial value of 20 to approximately 18, indicating increasing stability in public opinion distribution over time. However, the high structural constraints of regular lattice networks and the limited scope of opinion exchanges meant that agents could not achieve global consensus. Instead, they formed several stable opinion clusters—a characteristic feature of the bounded confidence model, and an illustration of opinion clustering dynamics. Since adoption thresholds were set to their highest values and the number of initial adopters was 0, no innovations were adopted by agents during the simulation (note that both adoption dynamics graphs in Fig 11 remained at zero). This outcome confirmed that the simulation's sole focus was on opinion dynamics, free of any influence from adoption processes.

The simulation results validated the downward compatibility of the BCAT model, demonstrating its ability to accurately replicate the opinion dynamics model under specific parameter configurations. The observed public opinion evolution entailed the formation of opinion clusters, the contraction of attitude distributions, and the difficulty of achieving global consensus—all consistent with theoretical predictions associated with a standard bounded confidence model. Further, the complete absence of adoption dynamics confirmed the model's separation of opinion and adoption processes.

As shown in Fig 12, the BCAT model is downward compatible with the adoption threshold model, which focuses on innovation diffusion. Parameters were set to eliminate the impact of opinion dynamics. Agent attitudes were initially set to 100 by configuring avg-of-attitudes at 100 and std-of-attitudes at 0, thus ensuring that all agents held identical positive opinions about the innovation in question, and removing the need for opinion exchanges. The opinion dynamics mechanism was disabled by setting the bounded-confidence parameter to 0. This allowed the simulation to isolate the adoption process so that all dynamics were driven solely by adoption thresholds.

The adoption process was governed by avg-of-thresholds (= 20) and std-of-thresholds (= 10), which respectively defined agent adoption mean and variability. This distribution created a range of thresholds across the agent population, representing diverse sensitivities to peer influence. The number of initial adopters (no-of-pioneers) was set to 3, introducing a small critical mass to initiate the diffusion process. Network structure was set to regular lattice due to the configuration of network-type = SWN and rewiring-probability = 0.0, ensuring that agents were only connected to a fixed set of neighbors in a highly structured manner.

During the simulation, agents evaluated whether the proportion of adopting neighbors exceeded their individual adoption thresholds. If a threshold was exceeded, the agent adopted the innovation. Adoption dynamics unfolded rapidly over the course of 50 ticks. Simulation data revealed a steep increase in the number of adopters within the first few ticks, reaching a critical point at which the adoption process accelerated significantly. This phenomenon, which is consistent with standard adoption threshold models, reflects the critical mass that drives innovation diffusion. All agents adopted the innovation by the end of the simulation. The adoption dynamics graph depicted a characteristic S-curve, with slow initial growth followed by a rapid increase and eventual saturation. The new adopter dynamics graph showed that the highest number of new adopters emerged at the critical point, in agreement with theoretical predictions from adoption threshold models.

The simulation validated the downward compatibility of the BCAT and adoption threshold models by demonstrating a capability to replicate standard diffusion dynamics. The complete exclusion of opinion dynamics ensured that the

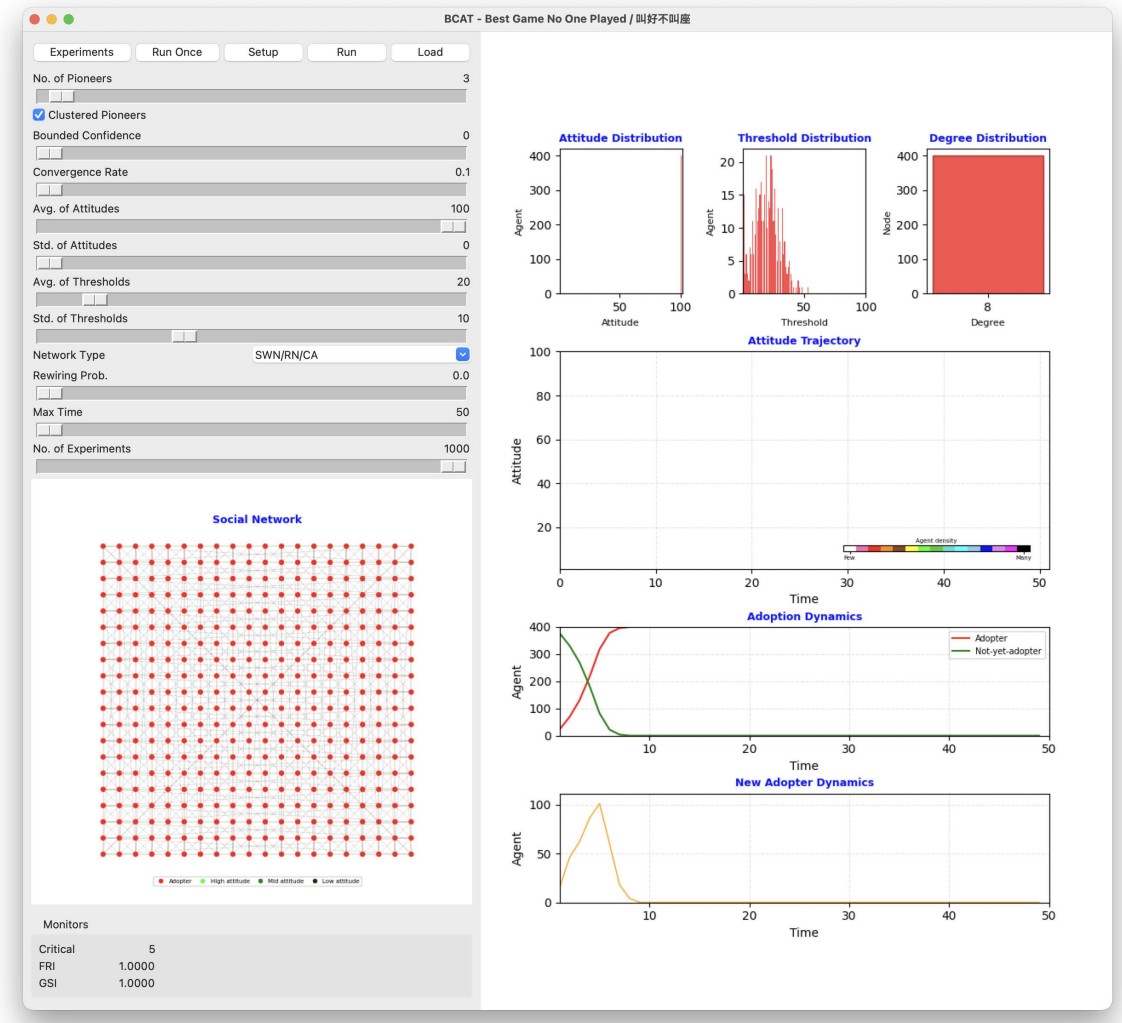

**Fig 12. BCAT model configured as an adoption threshold model, with opinion dynamics removed by uniform agent attitude parameters (avg-of-attitudes = 100, std-of-attitudes = 0).** Simulation consists of a regular lattice network structure with 400 agents and 1600 edges, with each agent connected to an average of 8 neighbors. Adoption process is governed by threshold distribution (avg-of-thresholds = 20, std-of-thresholds = 10). Results indicate complete adoption by all agents within 50 time ticks, highlighting the influence of critical mass, network structure, and threshold heterogeneity on innovation diffusion.

results were driven solely by the adoption thresholds, highlighting the separability of the two processes within the BCAT framework.

## Discussion and conclusion

Our mixed-integration BCAT model and sensitivity analysis results make three contributions to the computational social and management science literature. From a social perspective, the proposed model provides insights into collective behaviors, offering possible reasons why certain innovative products and ideas receive approval from experts and potential supporters, yet fail to gain widespread acceptance. Such insights may be useful for policy makers interested in social

advocacy campaigns. Second, from an economic perspective the model may assist corporate marketers wanting to identify factors that maximize new product adoption or minimize resistance, especially for controversial and exceptionally innovative products. Last, from an academic perspective, the BCAT model addresses an "innovation management gap" between positive initial attitudes and subsequent adoption or rejection behaviors. We believe the model can serve as a stimulus for interdisciplinary research in topic areas such as computational management science, innovation diffusion, social networks, and strategies for promoting innovative ideas and products. Further, sensitivity results across four kinds of theoretical networks offer valuable insights for optimizing innovative product or controversial policy success. Decision makers can adjust the five primary parameters to best influence public adoption behaviors.

Our mechanism decomposition experiments explain why some products with favorable average attitudes fail to achieve widespread adoption. The opinion clustering channel acts as an initial screen that reduces the potential adopter pool independently of threshold-based dynamics. Note that bounded confidence dynamics partition populations into stable attitude groups, some settling permanently below the adoption eligibility threshold. However, the effect of user-friendly testimony acts as a built-in corrective: when the adoption cascade propagates, adopting agents pull bounded confidence-reachable neighbors above the attitude threshold, largely neutralizing the opinion clustering gap. In contrast, the coordination failure channel depends on threshold levels when serving as the dominant driver of adoption gaps across all configurations, a decomposition suggesting distinct intervention strategies. Reducing opinion-driven gaps requires the targeted persuasion of agents found in borderline attitude clusters, while the reduction of coordination-driven gaps requires lowering effective adoption thresholds or increasing the spatial concentrations of early adopters. The two strategies operate on different levers, and are therefore complementary rather than substitutable.

In principle, several competing theoretical frameworks explain the opinion–adoption gap discussed in this study. Positioning the BCAT model's contribution requires a systematic comparison of the mechanisms and predictions generated by each framework. Three alternatives deserve special consideration: rational herding and information cascades [60], network externalities and coordination games [61], and global games with heterogeneous thresholds [62]. Each framework is capable of generating divergence between favorable opinions and low adoption rates, but the mechanisms, network topology effects, and adoption dynamics they predict differ from those of the BCAT model in ways that may be empirically discernible.

According to Banerjee's [60] information cascades model, adoption failure can be explained by the rational suppression of private information. Decision makers can observe the actions but not the private valuations of their predecessors, and rationally choose to imitate them when the inferential weight of observed actions exceeds that of their own private signals. Three predictions follow: first, adoption dynamics should exhibit a discontinuous jump at the cascade trigger point rather than a smooth cumulative curve, since no adoption can occur until the cascade condition is met. Second, the cascade is fragile—a single public counter-signal can reverse it instantaneously. Third, final adoption outcomes depend primarily on the action sequence of the earliest decision makers rather than on the population-level distribution of private valuations. The BCAT model generates different predictions for all three dimensions. Adoption follows a continuous S-shaped cumulative curve governed by the progressive updating of attitude values via bounded confidence communication (Figs 4 and 6a) instead of a step function. The process is more stable because persistent attitude updates do not depend on inferences from early movers. Further, the final GSI value is a systematic and monotonic function of five identifiable model parameters (Table 3). Fundamentally, social influences in the BCAT model operate by means of direct attitude transmission, with agents communicating their evaluations to one another and mutually adjusting their attitude values rather than drawing Bayesian inferences from observed choices (Algorithm 3). In the BCAT framework, word-of-mouth serves as genuinely persuasive communication rather than a signal indicating an unknown state. This distinction has direct implications for designing both empirical tests and marketing interventions.

Katz and Shapiro's [61] coordination game framework positions adoption failure within an external product valuation structure, with each agent's willingness to pay escalating in step with the total adopter pool, and with self-fulfilling

pessimistic beliefs concerning aggregate adoption. A key prediction is the direction of the network structure effect on the adoption gap. In coordination game models, agent decisions are based on beliefs about total adopter numbers—since sparser networks propagate less information about aggregate adoption levels, they should generate larger adoption gaps than denser networks. However, BCAT predicts the opposite: adoption failure is most pronounced in highly clustered, locally connected networks (regular lattice, small-world), where neighborhood constraints limit the influence of early adopters. It is least pronounced in high-connectivity networks (random, scale-free), where long-range links allow cascade propagation to bypass local cluster boundaries. The sensitivity analysis results shown in Table 3 are consistent with this prediction. The inhibitory effect of avg-of-thresholds on GSI is strongest in regular lattice and small-world networks, and moderately attenuated in random and scale-free networks. This confirms that network clustering amplifies adoption suppression in the BCAT framework—the opposite of the coordination game model's prediction. This network-topology signature is testable as a discriminating prediction factor between the two frameworks.

Morris and Shin's [62] global games framework introduces strategic uncertainty over higher-order beliefs. Adoption failure can occur even when all agents hold positive first-order beliefs about product quality, especially if agents are uncertain about positive opinions held by others about a product or idea. BCAT does not contain a corresponding mechanism. As demonstrated by the mechanism decomposition experiments, when $FRI = 1.0$ is guaranteed by construction and the adoption threshold is sufficiently low, adoption must occur without a residual adoption gap attributable to strategic belief uncertainty. The BCAT model's adoption failure channels (opinion clustering and threshold-based coordination failure) are also absent from the global games framework, which does not model continuous attitude evolution nor changes in communication-mediated opinions. Another distinction concerns practical measurability. The central parameter in the Morris-Shin framework (precision of an agent's private signals) is not directly observable in field settings. In contrast, the BCAT model's avg-of-thresholds and bounded-confidence parameters have direct behavioral counterparts that researchers can attempt to measure using preference surveys or threshold elicitation experiments.

It is important to emphasize that the three alternative frameworks and the BCAT model are not mutually exclusive in terms of accounting for any instance of adoption failure. In actual markets it is possible for features from all four mechanisms to be exhibited simultaneously. The theoretical value of specifying distinguishing predictions is tied to supporting future empirical efforts to identify which mechanism is dominant in a given market context. This identification process cannot be addressed without first establishing what each theory predicts, and where predictions converge or diverge.

Other factors distinguish BCAT from two classes of coupled dynamical models that researchers have applied to adoption and diffusion phenomena. Voter models with inertia [63] introduce a persistence parameter resulting in agents being less likely to switch opinions. While these models are capable of generating metastable heterogeneous states, they operate on binary opinion spaces and lack the continuous attitude scale and bounded confidence communication mechanism that allow BCAT to produce graded opinion clusters. Whereas two connected voter model agents can always influence each other regardless of attitude distance, communication in the BCAT model only occurs within the bounded confidence threshold, thus creating the selective interaction structure that drives opinion fragmentation. SIS/SIR epidemic frameworks model adoption as a contagion process in which contact with an adopter "infects" non-adopters probabilistically. While these models naturally generate S-shaped adoption curves and can incorporate heterogeneous transmission rates across network topologies, they treat adoption as a purely mechanical transmission event without an underlying attitude dimension. In contrast, BCAT requires agents to hold positive attitudes and exceed an adoption threshold before adopting, thereby producing a two-channel opinion–adoption gap that single-process contagion models cannot replicate.

However, real-world applications of the BCAT model require a nuanced understanding of how individual parameters influence opinion and adoption dynamics across different network structures. First, avg-of-thresholds is an essential determinant of adoption speed and scope, and decreasing it can significantly enhance adoption likelihood. A simple example of a method to reduce this factor is the offering of free trials or discounts aimed at reducing psychological barriers and encouraging consumer experimentation. Companies launching new technologies can provide demo versions or

limited-time offers to ease adoption-related anxiety. In policy contexts, pilot programs showcasing the low-risk and tangible benefits of controversial ideas can mitigate public resistance. This approach is particularly effective in regular lattice and small-world networks dominated by localized interactions, and where lower avg-of-thresholds values have the potential to initiate early adoption clusters capable of influencing broader networks.

A clear example of the role of avg-of-thresholds in suppressing adoption cascades despite favorable opinion climates is the commercial trajectory of the Palm handheld computer, introduced in 1996 [30]. A small population of technology enthusiasts rapidly adopted Palm devices. Their attitude values were strongly positive, and their mutual reinforcement resulted in a spatially concentrated early adopter cluster. However, mainstream consumers in that era operated with substantially higher adoption thresholds. Most individuals required visible evidence of widespread peer adoption before committing to an unfamiliar and relatively expensive device or device category, and endorsements from a few technically knowledgeable acquaintances were considered insufficient. The social network characteristics relevant to mass-market consumer electronics diffusion in the mid-1990s emphasized densely clustered local interaction groups, similar to the BCAT model's regular lattice topology. As illustrated in Fig 5, when avg-of-thresholds was set to 40 for a regular lattice network with otherwise favorable attitude parameters, agent attitudes converged steadily toward positive values (FRI approaching 1.0), while the adoption curve remained near zero. Only the pioneer cluster and a negligible percentage of immediate neighbors ever reached the adoption threshold. This configuration adequately captures most of the Palm scenario: a product with genuinely favorable expert opinion, an enthusiastic but spatially contained early adopter base, and a mainstream market whose threshold for peer-visible adoption evidence was never met. The simulation result thus underscores the importance of reducing average adoption thresholds as part of a successful marketing strategy, using approaches such as free trials, subsidized pricing, and highly visible demonstration programs.

Avg-of-attitudes also plays a central role in shaping a network's initial predisposition toward adopting a new product or idea. Marketers and policy makers can benefit from efforts to build favorable public sentiment prior to actual introduction—for example, executing targeted advertising campaigns that highlight the unique value or benefits of a product in order to establish a positive first impression among potential consumers. For controversial policies and ideas, circulating success stories or supportive data via media outlets can build trust and support. This strategy is especially impactful in scale-free networks in which hub nodes amplify positive attitudes, thus facilitating rapid network diffusion.

An example of the constraints imposed by baseline attitude level is the box office performance of the Taiwanese film *The Bold, The Corrupt and The Beautiful*. It received the Best Feature award at the 54th Golden Horse Film Festival in 2017, but attracted a small audience relative to its critical acclaim [31]. The film's reception revealed bifurcation in a relevant social network. The cinephile and film critic community, whose members made up the primary early audience and whose attitudes toward the film were strongly and uniformly positive, formed a high-attitude opinion cluster with limited connectivity to the broader moviegoing population. Mainstream audiences, whose initial attitudes toward dark, stylistically demanding dramatic films were neutral or mixed, represented a population with a substantially lower avg-of-attitudes baseline. When BCAT avg-of-attitudes is initialized at or below a midpoint, opinion dynamics governed by bounded confidence exchanges can gradually elevate attitudes within reachable clusters. However, the magnitude of this upward shift is constrained by both initial baselines and communication boundaries between clusters with large attitude distances. If the gap between a cinephile cluster's attitudes and the general audience's baseline exceeds the bounded confidence threshold, opinion influence from the critical community will likely fail to reach the mainstream audience—a pattern consistent with Figs 8c and 8d, where low initial avg-of-attitudes values produced adoption outcome plateaus largely insensitive to further parameter variation. The practical implication is to focus more on building favorable audience sentiment prior to release rather than rely on post-award publicity, especially when culturally specific or artistically challenging products are involved. This is particularly true within scale-free media network structures where hub nodes such as major media outlets and influential critics with broad established audiences can accelerate attitude convergence.

The influence of std-of-thresholds, which represents a target population's heterogeneity and adoption tolerance, is most pronounced in random and scale-free networks, where greater diversity promotes broader adoption patterns. Marketers can take advantage of this by tailoring communication strategies to meet the specific characteristics of different audience segments—for example, a tech company might emphasize product features to tech-savvy consumers, while highlighting ease of use for general audiences. Policymakers can similarly design targeted communication strategies for specific demographic sectors such as urban/rural or high-/low-income, thus improving the odds that diverse audiences believe their concerns are being addressed.

Bounded-confidence, a measure of tolerance for opinion differences, denotes stability in opinion exchanges, with enhanced opportunities for open and inclusive dialogue to improve product or idea acceptance. For products, creating online community forums where customers can interact and share experiences can support user engagement with sellers for openly sharing feedback. For policies, consultation sessions and opinion platforms help increase transparency and trust. Such strategies are particularly relevant for regular lattice and small-world networks in which localized interactions benefit from expanded dialogue, thus facilitating greater acceptance through mutual understanding.

A strong example of a bounded-confidence mechanism from the mid-2000s is the diffusion failure of the service-oriented architecture (SOA), a discrete services-focused software design approach [64]. SOA was championed by a community of technical software architects and systems integration specialists with deep understanding of distributed service design. However, the target adopter population primarily consisted of enterprise IT managers and C-suite decision-makers whose evaluative frameworks emphasized return on investment, implementation risk, and short-term operational continuity. The professional vocabulary and organizational concerns of these decision-makers were far removed from the technical discourses used by SOA proponents. In BCAT model terminology, two distinct population subgroups occupied attitude space and professional vocabulary positions separated by a distance that exceeded the bounded confidence threshold, thus blocking opinion exchanges between the technical pioneer community and the conservative enterprise management community. Neither group held an antagonistic attitude, but the channels required for communication were not open due to prevailing interaction constraints. As demonstrated in Figs 8a and 8b, a low bounded-confidence parameter severely limited the influential reach of positive-attitude agents, and instead generated stable opinion clusters in which technically enthusiastic subpopulations failed to affect the attitude distribution of a broader potential adopter pool. The collapse of SOA investment following the 2008 financial crisis is also consistent with the BCAT prediction—the event could be modeled as a sudden exogenous reduction in avg-of-attitudes across an enterprise decision-maker population, reflecting sharply reduced IT budgets and heightened risk aversion. Products whose diffusion depends on attitude transfer across professionally diverse communities are particularly vulnerable to such negative shocks.

Though less influential overall, std-of-attitudes occasionally contributes to product or idea adoption by nurturing opinion diversity. In localized networks (regular lattice, small-world), efforts to purposefully identify early adopter diversity can enrich opinion exchanges and attract greater attention. Examples include inviting diverse groups of testers to product launch activities to share reviews that highlight product adaptability across different user profiles. In terms of policy execution, this means including a broad range of representatives in discussions to ensure that multiple perspectives are heard, again with the goal of enhancing public engagement and acceptance.

Underlying network structure strongly influences the effects of parameter adjustments. Parameters such as avg-of-thresholds that determine adoption speed and extent are universally important regardless of network type, whereas the influences of parameters such as std-of-thresholds and bounded-confidence are more context-dependent, with increased sensitivity noted in high-connectivity networks (random, scale-free). Conversely, in networks where interactions are more localized (regular lattice, small-world), strategies that emphasize threshold reduction and attitude alignment are considered more effective.

In summary, decision makers interested in optimizing the potential success of innovative products or controversial policies need to strategically adjust the BCAT model's five core parameters of thresholds, positive attitudes, audience

diversity, opinion diversity and agent interaction—all actionable factors capable of significantly enhancing public adoption and engagement. When tailored to specific network structures, these approaches are capable of providing robust frameworks for influencing diffusion dynamics and achieving desired outcomes.

## Supporting information

**S1 File. Sensitivity analysis simulation data across four network topologies.** Raw simulation output data used for Table 3 and Figs 7–9, covering regular lattice, small-world, random, and scale-free networks ($N = 100{,}548$ total observations across four topologies).
(XLSX)

**S2 File. Mechanism decomposition experiment data (MD-A, MD-B, MD-C).** Raw simulation output for the three controlled experiments, covering regular lattice and small-world topologies (30,000 runs). Used for Fig 10 and our mechanism decomposition analysis.
(CSV)

**S3 File. Finite-size scaling experiment data ($N = 400$–2,500).** Per-run results and summary statistics for finite-size scaling experiments at $N = 400$ ($20 \times 20$), $N = 900$ ($30 \times 30$), $N = 1{,}600$ ($40 \times 40$), and $N = 2{,}500$ ($50 \times 50$), confirming the robustness of all qualitative phenomena across system sizes.
(CSV)

**S4 File. Python scripts for all reported statistical analyses and figures.** Includes analysis and visualization code for generating Table 3, Figs 7–10, and finite-size scaling results from the raw data shown in S1–S3 Files.
(ZIP)

## Acknowledgments

The authors wish to thank the anonymous reviewers for their constructive comments, which helped us improve manuscript clarity and rigor.

## Author contributions

**Conceptualization:** Chung-Yuan Huang, Sheng-Wen Wang.

**Data curation:** Chung-Yuan Huang, Sheng-Wen Wang.

**Formal analysis:** Chung-Yuan Huang, Sheng-Wen Wang.

**Investigation:** Chung-Yuan Huang, Sheng-Wen Wang.

**Methodology:** Chung-Yuan Huang, Sheng-Wen Wang.

**Project administration:** Chung-Yuan Huang, Sheng-Wen Wang.

**Resources:** Chung-Yuan Huang, Sheng-Wen Wang.

**Software:** Chung-Yuan Huang, Sheng-Wen Wang.

**Supervision:** Chung-Yuan Huang, Sheng-Wen Wang.

**Validation:** Chung-Yuan Huang, Sheng-Wen Wang.

**Visualization:** Chung-Yuan Huang, Sheng-Wen Wang.

**Writing – original draft:** Chung-Yuan Huang, Sheng-Wen Wang.

**Writing – review & editing:** Chung-Yuan Huang, Sheng-Wen Wang.

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
