## [Decision Letter · Decision Letter 0]

16 Mar 2026

PONE-D-26-01398Exploring the ‘Best Game No One Played’ Phenomenon Using A Mixed Opinion Dynamics and Innovation Diffusion ModelPLOS One

Dear Dr. Huang,

Thank you for submitting your manuscript to PLOS ONE. After careful consideration, we feel that it has merit but does not fully meet PLOS ONE’s publication criteria as it currently stands. Therefore, we invite you to submit a revised version of the manuscript that addresses the points raised during the review process.

Compliance with the recommendations of both reviewers is essential for final acceptance of the manuscript.

We look forward to receiving your revised manuscript.

Kind regards,

Krzysztof Malarz, D.Sc., Ph.D., M.Sc.

Academic Editor

PLOS One

Journal Requirements:

4. We note that your Data Availability Statement is currently as follows: “All relevant data are within the manuscript and its Supporting Information files.”

5. We note that Figures 2, 4, 5, 6, 10 and 11 in your submission contain copyrighted images. All PLOS content is published under the Creative Commons Attribution License (CC BY 4.0), which means that the manuscript, images, and Supporting Information files will be freely available online, and any third party is permitted to access, download, copy, distribute, and use these materials in any way, even commercially, with proper attribution. For more information, see our copyright guidelines: http://journals.plos.org/plosone/s/licenses-and-copyright.

1. You may seek permission from the original copyright holder of Figure(s) [#] to publish the content specifically under the CC BY 4.0 license.

Reviewers' comments:

Reviewer's Responses to Questions

**Comments to the Author**

1. Is the manuscript technically sound, and do the data support the conclusions?

Reviewer #1: Yes

Reviewer #2: Partly

2. Has the statistical analysis been performed appropriately and rigorously?

Reviewer #1: Yes

Reviewer #2: Yes

3. Have the authors made all data underlying the findings in their manuscript fully available?

Reviewer #1: Yes

Reviewer #2: Yes

4. Is the manuscript presented in an intelligible fashion and written in standard English?

Reviewer #1: Yes

Reviewer #2: Yes

5. Review Comments to the Author

Reviewer #1: The paper addresses an interesting question. The disconnect between opinion formation and adoption behavior is a well-recognized empirical phenomenon, and the idea of combining opinion dynamics with adoption threshold models to study it is natural and, to my knowledge, novel. The simulation results confirm the basic intuition that favorable attitudes are necessary but not sufficient for adoption, and provide useful comparative statics across network topologies.

My suggestions center on three themes. First, the theoretical framework could be presented more clearly, with greater emphasis on what makes the two models combine organically and where the key mechanisms diverge.

Second, the most interesting aspect of the paper—the wedge between opinion and adoption—deserves deeper development, both through richer empirical grounding and through comparison with alternative explanations.

Third, the paper would benefit from a discussion of how one might empirically distinguish the mechanisms proposed here from other theories that predict similar patterns. I elaborate these points in my report.

Reviewer #2: The paper introduces the BCAT (Bounded Confidence Adoption Threshold) model, an agent-based framework that couples opinion dynamics with a threshold-based diffusion mechanism. The central motivation is explaining why high-quality products with positive reception often fail to reach a commercial tipping point. This is an interesting and relevant goal.

The integration of these two modeling approaches is a significant conceptual contribution. The authors provide a thorough bibliographic review to justify this synthesis and include "downward compatibility" tests (Sections 4.4 and 4.5) that verify the model’s behavior under limiting conditions. While the multi-method sensitivity analysis is extensive, there are several methodological inconsistencies and theoretical inaccuracies that must be addressed before the manuscript is suitable for publication.

1) There is a fundamental discrepancy between the mathematical definitions and the simulation implementation. The authors cite the Deffuant et al. (2002) model, which is defined on a continuous opinion space with time-varying uncertainty. However, the BCAT implementation uses integer attitudes (1, 100) and a fixed bounded-confidence parameter. Methodologically, this makes the implementation a version of the HK (Hegselmann-Krause) model.

The authors must reconcile the theoretical framing in Section 2 with the actual simulation. It is important to note taht Deffuant dynamics (specifically regarding extremism) differ significantly from HK consensus/fragmentation patterns.

2) The manuscript leaves the update scheme ambiguous. Algorithm 3 suggests that at each tick, agents interact with one randomly chosen neighbor (asynchronous/sequential update). However, the overall simulation structure does not clarify if the adoption state is updated synchronously.

Given that update order qualitatively affects convergence in opinion dynamics (cf. Berenbrink et al., 2024), the authors must explicitly define and justify the choice of synchronous vs. asynchronous updating.

3) In Algorithm 3, the adoption threshold is treated as a percentage (0–100). In the standard diffusion literature (e.g., Watts 2004, Valente 1996), the threshold is strictly a fraction in (0 - 1)

While not a fatal error, this non-standard scaling complicates comparisons with existing literature and makes the interpretation of Table 2 (where "avg-of-thresholds = 40") potentially misleading. The authors should standardize these units or explicitly justify the deviation.

4) While the authors state that 1,000 runs were performed to ensure statistical significance (page 18), the reporting in Table 2 remains incomplete:

Algorithm Identity: The specific feature importance algorithm (e.g., Random Forest, SHAP, or XGBoost) used to generate the importance rankings is never identified.

Uncertainty Reporting: Table 2 provides point estimates only. The authors must include standard errors or confidence intervals to establish the statistical reliability of the parameter importance values.

Data Alignment: There appears to be a permutation error in the "Parameter Importance" section of Table 2; the "All" row values seem misaligned with the individual network columns. This requires urgent verification.

5) The model runs with a fixed size of N=400 agents. wich is a very small sample, especially in the analysis of cascades and phase transitions,

To ensure that the observed "Best Game No One Played" phenomenon is a robust collective behavior and not a stochastic artifact of a small system, the authors should provide a finite-size scaling analysis (e.g., testing N=1000, 5000 )

6) The authors frequently invoke "Critical Mass" (e.g., citing a value of 48 agents in Fig 6a) but fail to provide an operational definition within the model's formal structure.

It is unclear if these values are analytically derived or post hoc observations. The authors should define how "Critical Mass" is computed within the BCAT framework and whether it represents a first or second order phase transition in the adoption density.

7) The real-world cases (Palm, Taiwanese cinema) are used only as anecdotal motivation. While empirical calibration is difficult for abstract models, the authors should explicitly discuss the limitations of mapping the 1–100 attitude scale to real-world consumer data. Furthermore, they should briefly compare BCAT’s quantitative behavior with existing coupled models (e.g., voter models with inertia or modified SIS/SIR frameworks).

8) Reference "Gong et al. (2020)" has a mistake. The title field contains a fragment of a footnote regarding NetLogo source code modification. Correct or replace this bibliographic entry.

The BCAT model is an interesting conceptual framework for understanding market failures in high-quality products. However, the methodological issues listed above requiere a major revision.

6. PLOS authors have the option to publish the peer review history of their article (what does this mean?). If published, this will include your full peer review and any attached files.

Reviewer #1: No

Reviewer #2: No

---

## [Author Response · Author response to Decision Letter 1]

6 Apr 2026

The authors wish to thank the anonymous reviewers for their constructive comments, which helped us improve manuscript clarity and rigor. Please see the attached "Response to Reviewers.pdf" for our detailed point-by-point responses to all comments from the Academic Editor and both reviewers.

---

## [Decision Letter · Decision Letter 1]

28 Apr 2026

Using a mixed opinion dynamics and innovation diffusion model to explore the ‘best game no one played’ phenomenon

PONE-D-26-01398R1

Dear Dr. Huang,

We’re pleased to inform you that your manuscript has been judged scientifically suitable for publication and will be formally accepted for publication once it meets all outstanding technical requirements.

Kind regards,

Krzysztof Malarz, D.Sc., Ph.D., M.Sc.

Academic Editor

PLOS One

Additional Editor Comments (optional):

Reviewers' comments:

Reviewer's Responses to Questions

**Comments to the Author**

1. If the authors have adequately addressed your comments raised in a previous round of review and you feel that this manuscript is now acceptable for publication, you may indicate that here to bypass the “Comments to the Author” section, enter your conflict of interest statement in the “Confidential to Editor” section, and submit your "Accept" recommendation.

Reviewer #1: All comments have been addressed

Reviewer #2: All comments have been addressed

2. Is the manuscript technically sound, and do the data support the conclusions?

Reviewer #1: Yes

Reviewer #2: Yes

3. Has the statistical analysis been performed appropriately and rigorously?

Reviewer #1: Yes

Reviewer #2: Yes

4. Have the authors made all data underlying the findings in their manuscript fully available?

Reviewer #1: Yes

Reviewer #2: Yes

5. Is the manuscript presented in an intelligible fashion and written in standard English?

Reviewer #1: Yes

Reviewer #2: Yes

6. Review Comments to the Author

Reviewer #1: I have read the revised manuscript carefully. In my view, the authors have substantially addressed the main concerns raised in the previous round, and I do not see any remaining substantive issue that would require another revision.

The revision is much clearer about the conceptual rationale for combining bounded confidence opinion dynamics with an adoption threshold model. In particular, the authors now explain why bounded confidence is a natural complement to threshold-based adoption: unlike DeGroot-style averaging or voter-model dynamics, bounded confidence can generate persistent opinion clusters, which then interact meaningfully with the adoption threshold. The added discussion of the dual-filter mechanism and the adoption-to-opinion feedback loop also helps clarify what the combined BCAT model contributes beyond either component model alone.

The revised manuscript also provides a more convincing account of the wedge between favorable opinions and low adoption. The new mechanism decomposition is especially useful. By separating the opinion-clustering channel from the coordination-failure channel, the authors show that opinion clustering can reduce the pool of potential adopters in isolation, but that in the full model the user-friendly testimony effect can largely neutralize this channel when adoption cascades propagate. As a result, threshold-based coordination failure emerges as the dominant driver of the opinion–adoption gap in the main settings. This addition significantly strengthens the paper.

I also appreciate the new comparison with alternative explanations, including information cascades, network externalities/coordination games, and global games. This discussion helps clarify the distinctive predictions of the BCAT model and better positions the paper within the broader literature. The additional details on reproducibility, including source code, parameter files, supporting data, finite-size robustness checks, and a step-by-step protocol, are also appropriate and useful.

I have only minor editorial suggestions, which need not hold up publication. The manuscript is somewhat long and occasionally repetitive across the model-description, mechanism-decomposition, and discussion sections. The authors may wish to streamline some exposition before publication. In addition, a few comparisons with alternative theories could be phrased slightly more cautiously, since the precise predictions of those theories may depend on the exact model specification. These are matters of presentation rather than substance.

Overall, the revised manuscript is clearer, more rigorous, and more transparent than the previous version. I am satisfied with the authors’ responses and recommend acceptance.

Reviewer #2: I consider that the revised version of the manuscrip is suitable for acceptance and publication. Author have done a good revision

7. PLOS authors have the option to publish the peer review history of their article (what does this mean?). If published, this will include your full peer review and any attached files.

Reviewer #1: No

Reviewer #2: **Yes:** Marcelo N Kuperman

---

## [Editor Report · Acceptance letter]

PONE-D-26-01398R1

PLOS One

Dear Dr. Huang,

I'm pleased to inform you that your manuscript has been deemed suitable for publication in PLOS One. Congratulations! Your manuscript is now being handed over to our production team.

Kind regards,

on behalf of

Dr. Krzysztof Malarz

Academic Editor

PLOS One